# Global gain modulation generates time-dependent urgency during perceptual choice in humans

Peter R. Murphy[1,2], Evert Boonstra[1] & Sander Nieuwenhuis[1]

Decision-makers must often balance the desire to accumulate information with the costs of protracted deliberation. Optimal, reward-maximizing decision-making can require dynamic adjustment of this speed/accuracy trade-off over the course of a single decision. However, it is unclear whether humans are capable of such time-dependent adjustments. Here, we identify several signatures of time-dependency in human perceptual decision-making and highlight their possible neural source. Behavioural and model-based analyses reveal that subjects respond to deadline-induced speed pressure by lowering their criterion on accumulated perceptual evidence as the deadline approaches. In the brain, this effect is reflected in evidence-independent urgency that pushes decision-related motor preparation signals closer to a fixed threshold. Moreover, we show that global modulation of neural gain, as indexed by task-related fluctuations in pupil diameter, is a plausible biophysical mechanism for the generation of this urgency. These findings establish context-sensitive time-dependency as a critical feature of human decision-making.

[1] Institute of Psychology and Leiden Institute for Brain and Cognition, Leiden University, 2333 AK Leiden, The Netherlands. [2] Department of Neurophysiology and Pathophysiology, University Medical Center Hamburg-Eppendorf, 20246 Hamburg, Germany. Correspondence and requests for materials should be addressed to P.R.M. (email: murphyp7@tcd.ie).

D ecision-makers are adept at trading speed for accuracy to meet contextual demands[1–3]. If time is at a premium, decisions can be made quickly at the potential expense of accuracy. Conversely, the decision-making process can also be prolonged to facilitate additional information gathering and more accurate choices. By negotiating the speed-accuracy tradeoff (SAT) in this manner, behaving agents can maximize their rate of reward in environments with different temporal constraints[4–6].

Studies of decision-making support a broad class of models in which noisy evidence for each available choice is accumulated over time and a decision is made once the accrued evidence passes a criterion level, termed the decision bound[7–13]. Within this framework, one intuitive and parsimonious account of SAT asserts that speed emphasis is regulated by adjusting the level of the decision bound, such that less evidence is required for decision commitment in situations that demand faster decision-making. Aside from such situational or 'static' adjustments, the bound is typically assumed to be constant over the course of a single decision, thereby enforcing a fixed policy on commitment for a given decision-making context. For decades, models that invoked such a context-dependent, time-invariant bound have provided good fits to empirical SAT data (for example, refs 9,14–17).

Recently, convergent lines of research have brought the principle of time-invariance into question. Theoretical treatments have shown that a time-invariant decision policy is sub-optimal when the potential cost of continued deliberation grows over time[18,19]—as is the case, for example, when speed pressure is generated by means of a temporal deadline on choices[20]. In such settings, maximizing reward instead relies on dynamically lowering the evidence required for commitment as elapsed decision time increases. Additionally, recent primate single-unit recording studies indicate that a time-dependent, evidence-independent influence on the decision process is observable in the activity of neurons that reflect evolving decision formation[21–23], and moreover, that the strength of this time-dependency is highly sensitive to SAT manipulations. In particular, in both lateral intraparietal[22] and dorsal premotor[23] neurons, greater speed emphasis manifests in a combination of statically increased baseline firing rates (see also ref. 24) and a clear evidence-independent increase in firing rates with greater elapsed time. It has been proposed that these contextually-sensitive influences combine to form a neural urgency signal that expedites the

evolving decision process by driving it closer to a fixed threshold, which translates to a dynamic criterion on evidence[19,22,23,25–28].

While these findings have illuminated the mechanistic basis of SAT regulation in non-human primates, time-invariance remains a dominant assumption in the human decision-making literature[10] and recent empirical and model comparison reports have reinforced this stance[15,29–31]. Moreover, even in non-human primates, little is known about the neurophysiological source of urgency. In the present study, we address these outstanding issues. We first present convergent behavioural, electrophysiological and model-based evidence that human subjects do invoke an urgency signal with both static and time-dependent components to adapt to deadline-induced speed pressure. Next, we show that global modulation of neural gain, as reflected in task-related fluctuations in pupil diameter, is a plausible biophysical mechanism for the generation of urgency. Lastly, we report that human behaviour bears hallmarks of time-dependency even when speed pressure is mild and not a central feature of task design.

## Results

**Behaviour under deadline and free response.** In the first experiment that we report, twenty-one individuals made two-alternative perceptual decisions about the dominant direction of motion of a cloud of moving dots[32] (Fig. 1a). Each subject performed this task at a single level of discrimination difficulty that was tailored to their perceptual threshold, but under two levels of speed emphasis. In the 'free response' (FR) regime, subjects were under no external speed pressure, were instructed to be as accurate as possible, and were monetarily rewarded (penalized) for correct (incorrect) decisions. In the 'deadline' (DL) regime, the same task instructions and incentive scheme applied, with the addition of an especially heavy penalty—ten times that for an incorrect decision—if a decision was not made by 1.4 s after motion onset. The speed pressure imposed by this deadline led to faster median response times (RTs: DL = 0.70 ± 0.02 s; FR = 1.19 ± 0.07 s; $t_{20} = 8.1$, $P < 1 \times 10^{-6}$) and less accurate decision-making (DL = 77.8 ± 1.2%; FR = 86.8 ± 1.3%; $t_{20} = 6.6$, $P < 1 \times 10^{-5}$) relative to the FR regime (Fig. 1b).

Given the large penalty for missed deadlines, a sensible strategy in the DL regime is to always execute a response before the deadline[20]. Indeed, subjects missed the deadline on a median of only 0.14 ± 0.13% of trials, compared to 38.8 ± 4.0% of RTs

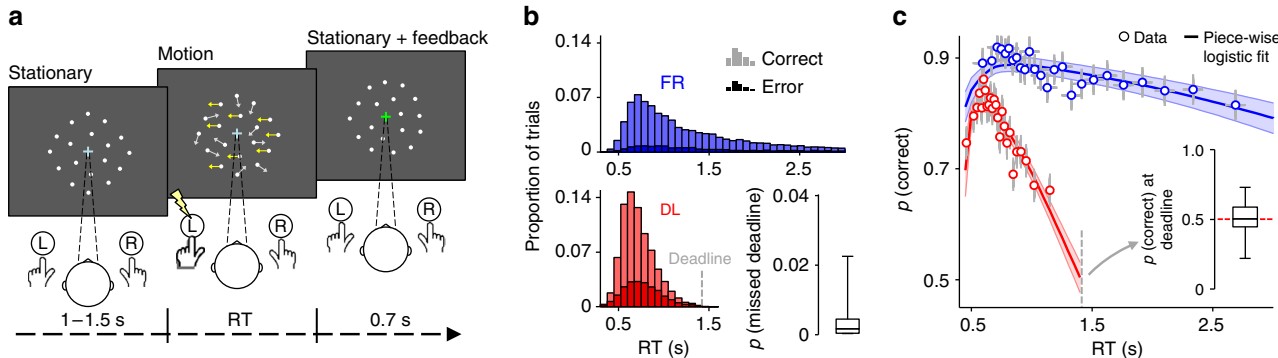

**Figure 1 | Perceptual task and associated behaviour.** (**a**) Schematic of a single-trial of the random dot motion task. (**b**) Subjects performed under 'free response' (FR) and 'deadline' (DL) conditions to manipulate speed pressure. Histograms depict pooled RT distributions from each condition. Box plot at lower right shows the sample median (centre line), interquartile range (box) and full range (whiskers) of proportion of missed deadlines in the DL condition. (**c**) Conditional accuracy functions. Points indicate mean accuracy of trials sorted by RT into 25 equal-sized bins and coloured lines show best fits of piece-wise logistic regressions to each subject's single-trial data. Error bars and shaded areas indicate ± s.e.m. of data points and regression lines, respectively. Box plot at inset shows the sample median (centre line), interquartile range (box) and full range (whiskers) of estimated accuracy at deadline in the DL condition.

exceeding this time in the FR regime (Fig. 1b). In accumulation-to-bound models of decision-making, this marked change in behaviour can primarily be achieved in two ways: by imposing a diminishing criterion on accumulated evidence as the deadline draws nearer, culminating in zero required evidence (and consequently, chance performance) around the time of the deadline; or, by lowering an otherwise static criterion sufficiently to ensure that effectively all decisions are made before the deadline, but are always based on the same quantity of accumulated evidence. While the latter mechanism predicts that the slope of the conditional accuracy function (CAF) relating accuracy to RT will be similar across speed emphasis regimes and that performance will generally not reach chance levels, the former predicts that the slope of the CAF will be substantially more negative in the DL regime and should arrive at approximately chance performance by the time of the deadline. Thus, empirical CAFs can, in principle, be used to arbitrate between different mechanistic accounts of SAT adjustment.

We employed single-trial logistic regression to estimate the shape of the empirical CAFs (see Methods; Fig. 1c). After accounting for a small percentage of fast inaccurate decisions, the estimated CAF slopes were negative in both the FR ($\beta = -0.35 \pm 0.07$, $t_{20} = -5.2$, $P < 1 \times 10^{-4}$) and DL ($\beta = -2.03 \pm 0.22$, $t_{20} = -9.3$, $P < 1 \times 10^{-8}$) regimes, but much more so in the latter (FR versus DL: $t_{20} = -8.3$, $P < 1 \times 10^{-7}$).

Moreover, using the DL regression fits to estimate accuracy at the time of the deadline revealed that this was not different from chance across subjects ($50.4 \pm 2.7\%$; one-sample $t$-test with $H_0 = 50\%$: $t_{20} = 0.2$, $P = 0.9$).

A negative CAF slope by itself does not necessarily imply time-dependency in the decision process; indeed, it should be expected whenever the strength of decision evidence fluctuates across trials, because trials with weak evidence will tend to be both slower and less accurate and thereby produce an asymmetry in correct and incorrect RT distributions[33]. Such evidence fluctuations can be due to variation in objective stimulus strength, but also to endogenous variation in attention or arousal[34]. We therefore examined the possibility that the CAF difference that we observed between the DL and FR conditions was simply caused by condition-related differences in arousal state. To do so, a second cohort of subjects performed the same motion discrimination task and we compared their CAFs on subsets of DL and FR trials that were precisely matched for pre-motion pupil size, a commonly-used metric of arousal and 'brain state' (see below). Even in this case of matched pupil-linked arousal, we observed a much more negative CAF slope under deadline ($t_{22} = -7.7$, $P < 1 \times 10^{-6}$; Supplementary Fig. 1).

Combined, the above observations are consistent with the adoption of a time-dependent decision policy in the DL regime. Two additional observations illuminate the nature of this policy

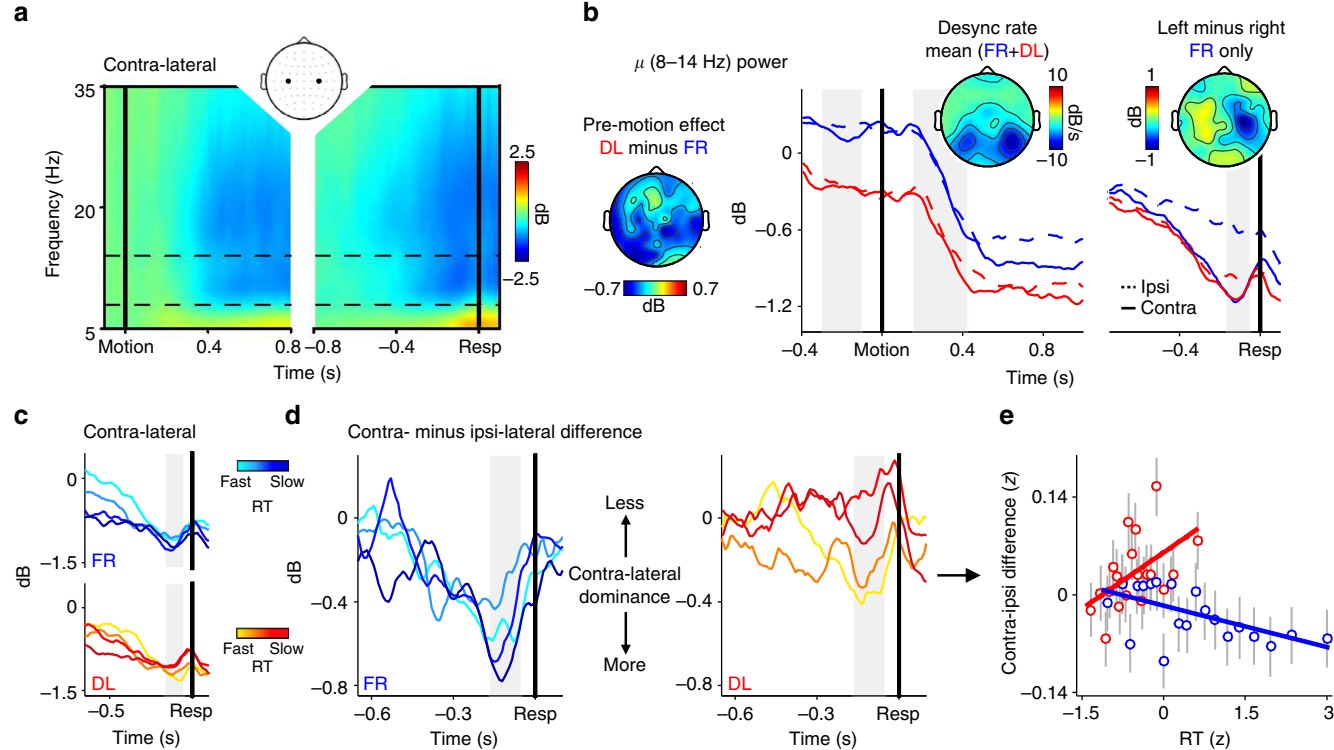

**Figure 2 | μ power tracks motor preparation and reveals urgency signatures under deadline-induced speed pressure.** (**a**) Time-frequency plot of oscillatory power over lateral motor channels, aligned to motion onset (left) and response (right). Plots show the trial-averaged change in power over the channel contra-lateral to the executed response on each trial, relative to a pre-motion baseline. (**b**) Onset and response-aligned μ (8–14 Hz) signals, separated by speed regime and lateralization relative to the executed response. Topographies at left, middle and right depict distribution of pre-motion effect of speed emphasis, stereotyped onset-evoked power decrease maximal over occipital scalp, and lateralization of μ power immediately prior to response execution in the FR condition, respectively. The level of contra-lateral desynchronization prior to response is highly similar across speed regimes, consistent with a common motor threshold. (**c**) Response-aligned μ signals contra-lateral to the executed response after sorting trials by RT into 4 equal-sized bins, separately for the FR and DL conditions. (**d**) Contra- minus ipsi-lateral difference waveforms, again after RT-sorting into 4 bins. Contra-lateral dominance prior to response execution decreases with slower RTs under deadline. (**e**) Scatterplot illustrating the linear relationships between RT and the contra-/ipsi-lateral μ difference for each speed regime. Points and error bars are mean ± s.e.m. of data that were z-scored within subjects, pooled across subjects and grouped into 20 bins; z-scoring was carried out across speed regimes to preserve main effects of speed emphasis. In **b**–**d**, shaded grey regions show measurement windows for scalp topographies and associated effects reported in text.

adjustment further. First, the smooth gradient of the right tail of the DL RT distributions and their associated CAF indicate that the time-dependent change in required evidence was gradual, not abrupt. Second, despite task difficulty being fixed across both speed emphasis regimes, peak decision accuracy across all RTs (as measured at the inflection point of the estimated CAFs) was reliably lower under deadline ($84.3 \pm 1.4$ versus $89.8 \pm 1.2\%$; $t_{20} = -4.0$, $P = 0.0008$). This suggests that, further to the time-dependent effect, additional speed emphasis was generated by a static, time-invariant lowering of the criterion on accumulated evidence which ensured that even fast decisions were less accurate in the DL regime.

**EEG motor preparation signatures of urgency.** Although these behavioural findings suggest that greater speed emphasis under deadline was achieved by a combination of static and time-dependent adjustments to the decision policy, they are not decisive about the mechanistic basis of these adjustments. One possibility is that the decision bound itself varies with speed emphasis and progressively collapses as the deadline approaches (for example ref. 35). Alternatively, the bound might remain fixed and an urgency signal could generate speed emphasis by providing additional input to each evidence accumulator. In an attempt to adjudicate between these competing mechanistic accounts of the static and time-variant influences on decision-making behaviour, we measured scalp EEG and examined motor preparatory activity via oscillatory power in the $\mu$ (8–14 Hz) frequency range. Previous studies have shown that the commonly observed decrease in $\mu$ power during decision formation reflects dynamic motor preparation that appears to be driven by the evidence accumulation process[36–38]. Here, we used effector-selective $\mu$ signals that were contra- and ipsi-lateral to the executed response as proxies for trial-by-trial preparatory activity in favour of the chosen and unchosen motion directions, respectively.

We observed that bi-lateral motor $\mu$ signals (Fig. 2a) were sensitive to speed emphasis in several distinct ways. First, there was a reliable effect of speed emphasis on $\mu$ power prior to motion onset such that pre-motion power was lower in the DL regime compared with the FR regime (mean $\beta = -0.102 \pm 0.044$, $t_{20} = -2.3$, $P = 0.03$; Fig. 2b, left). There was no main effect of lateralization (contra- versus ipsi-) during this period ($\beta = -0.019 \pm 0.015$, $t_{20} = -1.2$, $P = 0.2$), and no speed emphasis by lateralization interaction ($\beta = 0.016 \pm 0.021$, $t_{20} = 0.8$, $P = 0.5$). Thus, potentially indicative of a static urgency effect, greater speed pressure was accompanied by increased baseline motor preparation in both effectors. Topographic visualization of this pre-motion effect revealed that although foci of decreased power in the DL regime were apparent over bi-lateral motor channels, the effect also extended over posterior scalp. However, the effect over lateral motor areas remained marginally significant even when posterior 8–14 Hz activity was included as a co-variate ($\beta = -0.068 \pm 0.038$, $t_{20} = -1.8$, $P = 0.09$), suggesting that this motor effect was at least partially distinct from the more posterior effect.

Next, we turned to $\mu$ power prior to response execution in order to examine effector-specific motor preparation at decision commitment (see Methods for rationale behind selectively focusing on this measurement period). Consistent with previous findings[36–38], there was a lateralization in pre-response $\mu$ power: a greater decrease was present in contra-lateral rather than ipsi-lateral channels, reflecting greater motor build-up in favour of the ultimately executed response (Fig. 2b, right). Accordingly, a main effect of lateralization was observed in a statistical model with lateralization and speed emphasis regime as factors ($\beta = -0.120 \pm 0.044$, $t_{20} = -2.7$, $P = 0.013$). However, this

effect was also accompanied by a main effect of speed emphasis ($\beta = -0.089 \pm 0.034$, $t_{20} = -2.6$, $P = 0.016$) and a lateralization by speed emphasis interaction ($\beta = 0.095 \pm 0.025$, $t_{20} = 3.8$, $P = 0.001$). *Post-hoc* models revealed that while the expected contra/ipsi lateralization was clearly apparent in the FR regime ($\beta = -0.129 \pm 0.043$, $t_{20} = -3.0$, $P = 0.007$), it was not reliable under deadline ($\beta = -0.040 \pm 0.034$, $t_{20} = -1.2$, $P = 0.3$). Moreover, the pre-response ipsi-lateral signals representing motor preparation for the unchosen alternative were of significantly lower power in the DL relative to the FR regime ($\beta = -0.093 \pm 0.031$, $t_{20} = -3.0$, $P = 0.008$), whereas the contra-lateral signals reached a highly similar level ($P = 0.9$). All of these effects were also present when only subsets of RT-matched trials from each speed emphasis condition were analysed (Supplementary Fig. 2).

The stereotyped level of pre-response contra-lateral $\mu$ power suggests that the level of motor preparation required to execute a response was the same across both speed emphasis regimes. We also observed that this metric was invariant to RT within each regime (FR: $\beta = -0.012 \pm 0.014$, $t_{20} = -0.8$, $P = 0.4$; DL: $\beta = 0.025 \pm 0.014$, $t_{20} = 1.8$, $P = 0.09$; Fig. 2c). To the extent that pre-response $\mu$ may provide a proxy for the level of the decision bound, this pattern of findings is consistent with a fixed bound across speed emphasis regimes and decision times and therefore argues against the notion that speed emphasis is generated by bound adjustment. Instead, the lower $\mu$ power that was evident in the DL regime during the pre-motion period, and in the ipsi-lateral signal at the time of commitment, might plausibly reflect an urgency signal that provides an additional source of input to the evidence accumulation process.

In the above respects, our findings are consistent with recent studies of the neural basis of SAT regulation in non-human primates[22,23]. In a further analysis, we investigated whether, as in these studies, there was a time-dependent component to the urgency signal. In non-human primates, time-dependency in the neural urgency signal manifests as a building, common increase in the firing rates of neurons reflecting evidence accumulation for both the chosen and unchosen task alternatives[21–23]. In the case of our pre-response motor preparation signals, this time-dependent increase in common activation (or put differently, the time-dependent decrease in the difference between accumulators) should translate into a diminishing contra/ipsi lateralization with increasing RT (see Methods). Accordingly, we observed a speed regime by RT interaction ($\beta = 0.055 \pm 0.023$, $t_{20} = 2.4$, $P = 0.026$; Fig. 2d,e) in a model that examined the effect of these factors on pre-response $\mu$ lateralization. *Post-hoc* tests indicated that although there was no reliable relationship between $\mu$ lateralization and RT in the FR regime ($\beta = -0.020 \pm 0.013$, $t_{20} = -1.6$, $P = 0.1$), the strength of lateralization decreased as predicted for slower RTs in the DL regime ($\beta = 0.020 \pm 0.009$, $t_{20} = 2.2$, $P = 0.038$). This finding supports the hypothesis that, in addition to the static pre-motion effect described earlier (Fig. 2b, left), greater speed emphasis under deadline was generated by a time-dependent urgency signal that increased in magnitude as the deadline drew nearer.

**Drift diffusion modelling corroborates urgency account.** In light of this combined behavioural and electrophysiological support for static and time-varying urgency as mechanisms for generating greater speed emphasis, we proceeded to verify that a computational model that incorporates these features can account for the observed behavioural data. In our model, a decision is made when one of two anti-correlated evidence accumulators reaches a fixed decision bound, and the accumulators are subject to the same additive, time-varying urgency signal that is free to

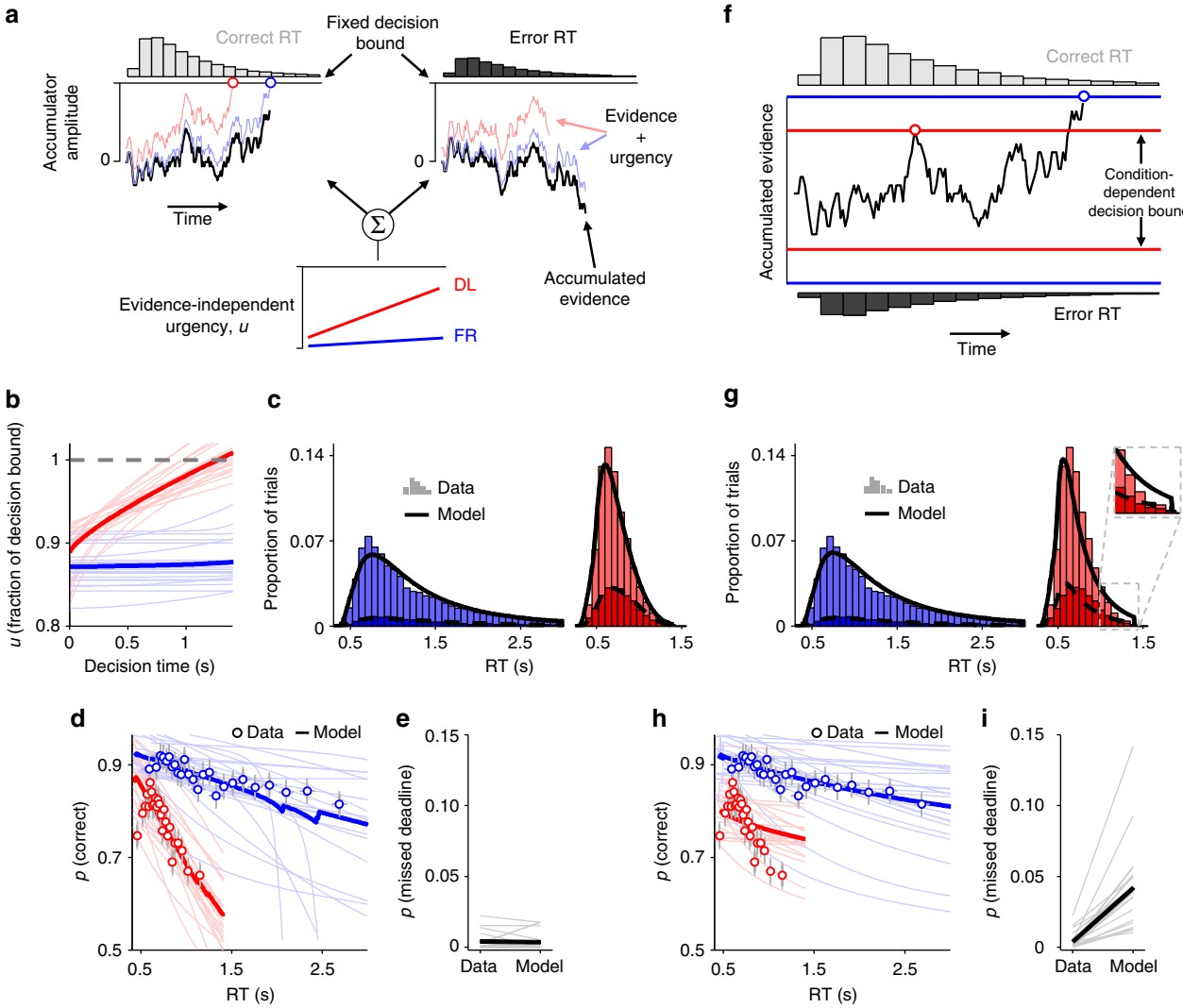

**Figure 3 | Comparison of diffusion model fits with and without urgency. (a)** Schematic representation of the drift diffusion model (DDM) with urgency and fixed decision bounds. **(b)** Urgency signals, modelled as a logistic function, derived from fits of the urgency DDM to behaviour. Lighter lines indicate fits for individual subjects; darker lines indicate group-averages. **(c)** Observed and fitted RT distributions (histograms and lines, respectively), pooled across subjects. **(d)** Conditional accuracy functions from urgency DDM fits. Points indicate mean accuracy of trials sorted by RT into 25 equal-sized bins (error bars indicate ± s.e.m.); dark/light lines depict group-average/single-subject model fits, respectively. **(e)** Observed and fitted proportion of missed deadlines. **(f)** Schematic of the regular DDM without urgency. **(g–i)** Fitted RT distributions **(g)**, conditional accuracy functions **(h)** and proportion of missed deadlines **(i)** derived from fits of the regular DDM to behaviour.

take different shapes across speed emphasis regimes (Fig. 3a; Methods). Without urgency, this model reduces to the popular drift diffusion model (DDM) in which a single accumulation process plays out between two opposing bounds[10] (Fig. 3f).

We fit a variety of models with urgency to the data and found that the best-fitting model allowed both the shape of the urgency signal and a non-decision time parameter to vary across speed emphasis regimes (Supplementary Table 1). The rate of evidence accumulation, known as the drift rate, was allowed to vary across trials in this model (Wilcoxon signed-rank tests on BIC differences: $P < 0.1$ for all pair-wise comparisons of otherwise identical model variants with and without drift rate variability), but mean drift rate and the magnitude of this between-trial variability were fixed across speed regimes. This best-fitting urgency model fit the observed RT distributions well (Fig. 3c) and was able to reproduce the key qualitative behavioural effects of increased speed emphasis under deadline (Fig. 3d,e): the more negative CAF slopes; the very low proportion of missed deadlines; and, in most subjects, the tendency toward near-chance

performance at the time of the deadline. Notably, there were effects of speed emphasis on both the baseline offset and the time-varying shape of the fitted urgency signals, corresponding to greater static and time-dependent urgency under deadline, respectively, and the shape of the fitted urgency signals was highly consistent across subjects (Fig. 3b). Moreover, non-decision times were found to be marginally faster in the DL regime compared with the FR regime (Supplementary Table 2).

By contrast, the standard DDM with condition-dependent but time-invariant decision bounds (Fig. 3f) provided a considerably poorer fit to the observed data (Fig. 3g–i; Supplementary Table 1; Wilcoxon signed-rank tests on BIC differences: $P < 0.001$ for all pair-wise comparisons of urgency models with their standard DDM counterparts). This poor fit stems from the fact that the standard DDM is incapable of generating a more negative CAF slope, to the extent required here, without also increasing the proportion of missed deadlines. Its main mechanism for lowering the slope of the CAF is to increase the between-trial variability in drift rate[33,34]; but, this produces a relative increase in the

proportion of trials that have a near-zero drift rate, which are less likely to reach the decision bound before the deadline. As a consequence, in our standard DDM fits, leaving between-trial variability in drift rate free to vary across speed emphasis regimes did not even yield an increase in goodness-of-fit (Supplementary Table 1). Thus, quantitative model comparisons corroborated the presence of urgency with time-dependency in the decision process under deadline.

Using the closed-form function for the urgency signal in the above model fits (equation 4), we also approximated the optimal, reward-maximizing shape of time-dependent urgency on our task for a representative set of remaining model parameters (Supplementary Fig. 3). This optimal urgency signal required a fast transition from a flat early portion to a steep deflection toward the decision bound closer to the deadline (cf. refs 20,30) that is qualitatively very different from the gradual, approximately linear urgency signals that subjects in the current study appeared to implement. As such, although our subjects responded to deadline-induced speed pressure by adjusting their decision policies in a time-dependent fashion, they failed to do so optimally. Interestingly, when the urgency signal was further constrained to be strictly linear in the optimality calculations, its reward-maximizing trajectory was matched much better by the fitted signals derived from the observed data (Supplementary Fig. 3). Combined, these observations may point to limitations of the neural mechanisms responsible for urgency generation (see Discussion).

**Pupillometry highlights gain modulation as source of urgency.** While the above findings describe the effects of urgency on behaviour and cortical signatures of decision-related motor preparation, they do not shed light on the neural origins of the urgency signal. Theoretical accounts have identified gain modulation, which affects the responsivity of both excitatory and inhibitory neural connections, as a potential mechanism for generating urgency in the brain[19,26,27,39,40]. However, this possibility has not been tested empirically. In the second experiment, we investigated whether global, brain-wide gain modulation, as indexed by pupil diameter, may be implicated in the injection of urgency into the decision process. Under constant luminance, changes in pupil diameter have been linked to the activity of diffusely-projecting neuromodulatory systems, in particular the locus coeruleus-noradrenergic (LC–NA) system[41–43], that are thought to control global neural gain[44–47].

A second cohort of twenty-three subjects (whose CAFs are already reported in Supplementary Fig. 1) performed the motion discrimination task optimized for measurement of decision-related changes in pupil diameter. We first examined the effect of speed emphasis on unbaselined pupil diameter prior to motion onset, which has previously been used as a proxy for 'tonic' fluctuations in neural gain[45,48]. Consistent with a static increase in gain under greater speed pressure, this metric was larger in the DL regime than in the FR regime ($t_{22} = 6.9$, $P < 1 \times 10^{-6}$; Fig. 4a).

Next, we examined the effect of speed emphasis on evoked, 'phasic' pupil dilations after motion onset and whether this effect interacted with RT, as expected of an urgency signal with a strength that depends on elapsed decision time. We observed a main effect of speed regime on trial-by-trial pupil dilation magnitude, driven by larger dilations in the DL regime than the FR regime ($\beta = 0.143 \pm 0.048$, $t_{22} = 3.0$, $P = 0.007$; Fig. 4b). Moreover, there was a significant speed regime by RT interaction ($\beta = 0.189 \pm 0.040$, $t_{22} = 4.7$, $P < 1 \times 10^{-4}$; Fig. 4c,d). *Post-hoc* tests revealed that while no reliable relationship existed between dilation magnitude and RT in the FR regime ($\beta = -0.022 \pm 0.020$, $t_{22} = -1.1$, $P = 0.3$), pupil dilations were

larger for slower RTs in the DL regime ($\beta = 0.085 \pm 0.017$, $t_{22} = 5.0$, $P < 1 \times 10^{-4}$). These effects were present across a broad range of both stimulus- and response-aligned measurement windows (Supplementary Fig. 4).

We next sought to identify the most likely shape of the neural input to the pupil system during decision formation by combining linear systems analysis with formal model selection. In accordance with recent reports[49,50], the trial-related input to the pupil system was modelled as a linear superposition of three temporal components: a transient at motion onset, a transient at response, and a sustained component throughout the intervening period of decision formation, each convolved with a pupil impulse response function[51]. Using this approach, we then compared the goodness-of-fit of a variety of models in which the shape of the sustained decisional component varied (Fig. 5a; Methods). The model that best fit the pupil data from the FR regime was one in which the input to the pupil system maintained a constant amplitude throughout the decisional period (a 'boxcar'), irrespective of how long the decision took to be made (Fig. 5b). In contrast, the best fit to the DL data was provided by a model in which input strength ramped up monotonically with elapsed decision time (Fig. 5c). The latter finding reflects a truly time-dependent increase in the neural input to the pupil system in the DL regime, thus supporting the hypothesis that the gain of neural processing increased with elapsed time under speed pressure. Additionally, the boxcar and linear up-ramp remained the best-fitting models of the FR and DL data, respectively, across a wide range of different parameterizations of the pupil impulse response function (Supplementary Fig. 5).

In both speed emphasis regimes, each of the three modelled temporal components contributed significantly to the measured pupil time series (Fig. 5d,e). Hence, we also tested whether the observed relationship between pupil dilation and RT in the DL regime (Fig. 4c,d) was fully captured by the RT modulation inherent in the ramping decisional component of the associated best-fitting model, or if the onset and response components also contributed to this effect. In model variants that included additional terms representing the parametric modulation of each temporal component by RT, neither modulated term for the onset or response components contributed consistently to the DL pupil time series (Effect size for parametrically modulated onset term $= -4.5 \pm 3.2$, $t_{22} = -1.4$, $P = 0.2$; Effect size for parametrically modulated response term $= 2.7 \pm 3.2$, $t_{22} = 0.8$, $P = 0.4$). This suggests that the dilation/RT relationship in the raw DL data reflects a time-dependent modulation of input to the pupil system that was specific to the period of decision formation.

**Global gain modulation alone produces urgency effects.** To build on these pupillometric observations, we next verified that a combination of static and time-dependent changes in global gain is capable of producing the qualitative effects of deadline-induced speed pressure on both overt behaviour and decision-related neural dynamics. We modelled global gain modulation as a change in the slope of the input-to-output transfer function of a simple neural network that incorporates basic principles of neural computation[12] (Fig. 6a,b). Informed by our pupillometric results, gain was fixed at a low level throughout a trial in the FR regime but subject to static and time-dependent increases in the DL regime (Fig. 6c). All other model parameters, aside from non-decision time, were fixed across regimes (Methods).

When fit to the pooled behaviour of the cohort of subjects from the first experiment reported above, this model successfully reproduced all of the key qualitative effects of deadline-induced speed pressure on both behaviour (Fig. 6d,e; proportion of missed

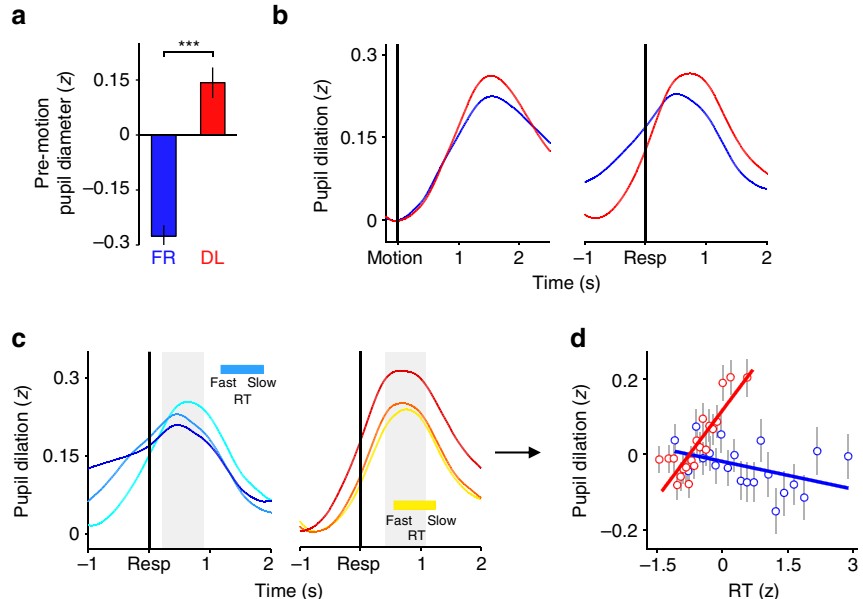

**Figure 4 | Effects of deadline-induced speed pressure on pre-motion pupil diameter and evoked pupil dilation.** (**a**) Effect of speed regime on unbaselined pupil diameter, as measured during the period directly preceding motion onset. Error bars = s.e.m. *** = $P < 0.001$. (**b**) Evoked pupil dilation separated by speed regime and aligned to both motion onset (left) and response (right). Waveforms are relative to a pre-motion baseline. (**c**) Response-aligned pupil dilation after sorting trials by RT into 3 equal-sized bins, separately for the FR and DL conditions. (**d**) Scatterplot illustrating the linear relationships between RT and evoked pupil dilation for each speed regime. Points and error bars are mean ± s.e.m. of data that were z-scored within subjects, pooled across subjects and grouped into 20 bins; z-scoring was carried out across speed regimes to preserve main effects of speed emphasis. In **c**, shaded grey regions show measurement windows for effects shown in **d** and reported in text.

deadlines < 0.1%) and the dynamics of the evidence accumulation process. With respect to the accumulation dynamics, the activation time-series of the simulated accumulator units in the model displayed the two critical characteristics of static and time-dependent urgency that we observed under deadline in motor preparation signals in the human EEG: a baseline increase in activation during the pre-motion period (Fig. 6f); and, stronger common activation of both accumulators (reflected in a smaller difference between accumulators) at the time of commitment for slower decision times (Fig. 6g,h). These simulations suggest that global gain modulation is a plausible biophysical mechanism for generating static and time-dependent urgency in the brain.

It has recently been argued that, rather than relying on gradual evidence accumulation, decisions are determined by a more instantaneous estimate of the current sensory evidence combined with a growing urgency signal[19,23,28,52]. In our simple network model, such a regime can be approximated by constraining the effective time constant of accumulation ($\tau$) to be particularly short (Supplementary Methods). When we enforced this constraint, the model still provided a reasonable account of behaviour and accumulation dynamics (Fig. 6d,e,h, thin grey lines). Thus, evidence accumulation with a long time constant does not appear to be a necessary prerequisite for generating the data observed presently.

**Time-dependent urgency under mild speed pressure.** The signatures of urgency that we report above were observed in task contexts of high speed pressure. In a final set of analyses, we examined whether the same mechanism might also be invoked in situations where speed pressure is less severe. We re-analysed data from two experiments in which subjects again made motion discrimination decisions, but without any manipulation of speed emphasis. Instead, they performed under a deadline of 1.5 s at all times and there was no explicit penalty for missed deadlines. This task feature has been employed previously in studies of human

perceptual decision-making that were not designed to interrogate the mechanistic basis of SAT regulation (for example, refs 34,53).

In the first of our re-analysed studies[34] (Fig. 7a), subjects missed a low proportion of deadlines (median = 0.50 ± 0.16%), and their CAFs arrived at a mean accuracy level at the time of the deadline that was not different from chance across subjects (48.0 ± 3.9%; $t_{25} = -0.5$, $P = 0.6$). In the second study (unpublished; Fig. 7b), subjects performed under two difficulty levels and again missed very few deadlines (easy = 0.15 ± 0.13%; hard = 0.63 ± 0.15%). Moreover, despite the CAFs for each difficulty level being significantly different for almost the entire range of RTs, they converged to approximately chance accuracy at the deadline (easy: 51.9 ± 4.5%, $t_{20} = 0.4$, $P = 0.7$; hard: 46.2 ± 2.0%, $t_{20} = -1.9$, $P = 0.07$; paired-samples t-test for easy versus hard: $t_{20} = 1.3$, $P = 0.2$). As described previously, this repeatedly observed combination of few missed deadlines, strongly negative CAF slopes and chance performance around the time of the deadline is a hallmark of a time-dependent decision policy.

## Discussion

In models of decision-making, a common assumption is that the accuracy and timing of decision commitment are determined by a context-dependent but time-invariant criterion on accumulated evidence[7–10,16]. Theoretical considerations suggest that such a time-invariant policy is sub-optimal if the potential cost of continued evidence accumulation grows with elapsed decision time, as is often the case in decision-making contexts that place a premium on fast responding[18,20]. Yet, in support of the principle of time-invariance, recent reports have suggested that human decision-makers may fail to implement a dynamic, time-variant commitment policy that would yield higher reward rates in such settings[15,29,30]. In the present study, we describe strong, convergent evidence to the contrary. Through analysis of observed behaviour, computational modelling and scalp

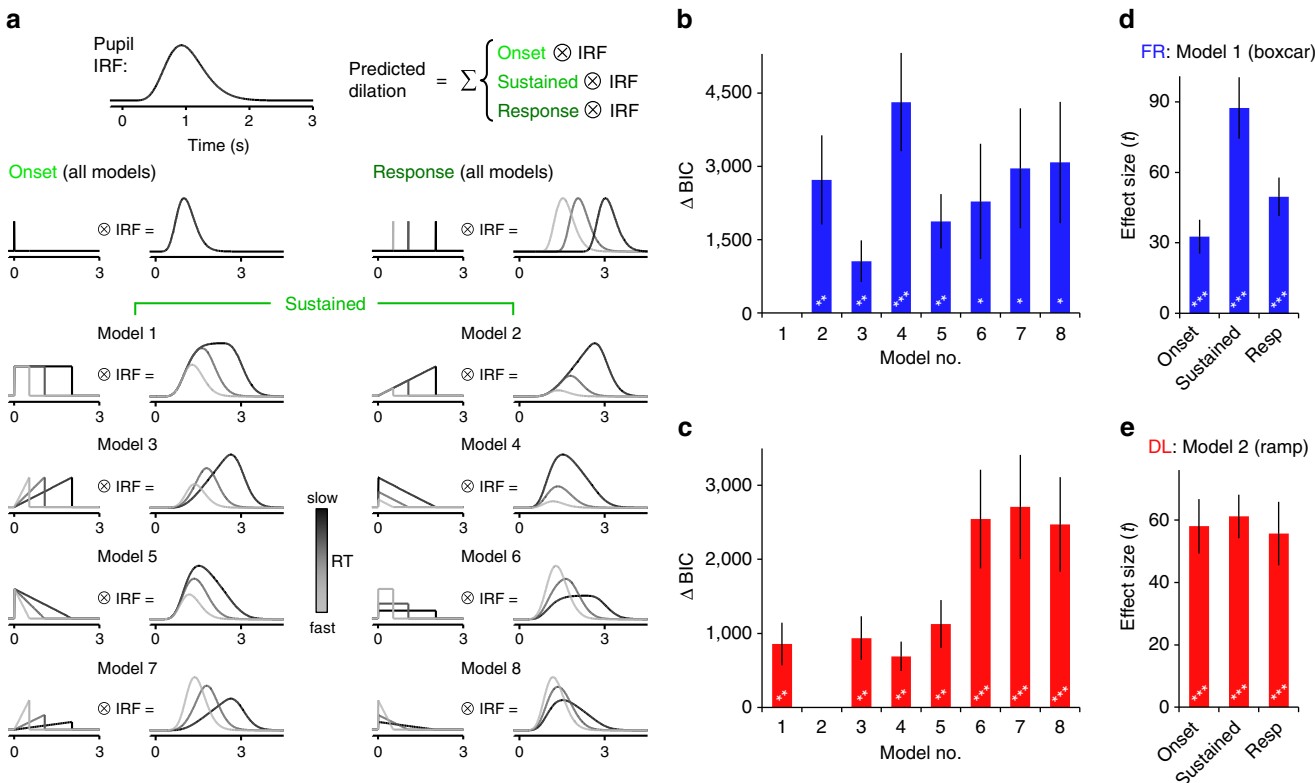

**Figure 5 | Identifying the shape of the neural input to the pupil diameter system during decision formation.** (**a**) Schematic depicting the analysis approach employed to parse the contributions of distinct temporal components to the observed pupil dilation waveforms and identify the likely shape of the input to the pupil system during decision formation. Three temporal components (onset, sustained, response) were convolved with a canonical pupil impulse response function (top left) and regressed onto the observed pupil diameter time series. The shape of the sustained decisional component differed across 8 candidate models and their goodness-of-fits were quantitatively compared. (**b,c**) BIC scores for each of the 8 models fit to the FR (**b**) and DL (**c**) data, relative to the winning model in each case (model 1 for FR; model 2 for DL). (**d,e**) Effect sizes of parameter estimates for each of the temporal components from the winning FR (**d**) and DL (**e**) models. Error bars = s.e.m. ***=$P < 0.001$, **=$P < 0.01$, *=$P < 0.05$.

electrophysiological and pupillometric findings, we show that human subjects are capable of adapting to deadline-induced speed pressure via a combination of static and time-dependent changes to their criterion on accumulated evidence.

Recent studies that applied quantitative model comparison techniques to multiple behavioural datasets provided some support for the presence of a time-dependent influence on the decision process of highly-trained monkeys, but little evidence for time-dependency in mostly naïve human subjects[15,31]. By contrast, we observed clear support for model variants with strong time-dependency in humans that, aside from brief initial training sessions, had no prior experience with the imperative task. How might this discrepancy in findings be explained? One likely contributing factor is differences in the nature of the speed pressure created by the various decision-making contexts in question. In our task, the heavy punishments levied for missed deadlines created strong, time-sensitive speed pressure that was likely sufficient to mitigate the bias toward accurate over reward-maximizing behaviour that human subjects can display can display in choice RT settings[4,54]. On the other hand, this may not have been the case in previous studies that imposed only small, implicit penalties for slow responses (in the form of foregone rewards; for example, ref. 30), or did not provide performance-related incentives at all (for example, ref. 31).

It is also possible that mild time-dependency was present in previous investigations but not identifiable in model fits to behaviour. Specifically, popular time-invariant sequential sampling models can include variability parameters that produce similar behavioural effects as moderate time-dependent changes

in the decision policy[7,33,34], potentially rendering the two indistinguishable via model comparison alone. In our case, targeted analysis of overt behaviour, measured in contexts of both strong and mild deadline-induced speed pressure, revealed signatures of time-dependency that cannot, in principle, be produced solely by variability parameters. These behavioural patterns are driven by a small percentage of trials with RTs close to the deadline and in many cases may exert a negligible influence on likelihood estimates commonly used for model fitting, but can nonetheless be highly informative when attempting to arbitrate between competing mechanistic accounts. Thus, future investigations of time-dependency in the decision process might benefit from invoking a combination of formal model comparison and assessment of such behavioural trends.

A third possibility is that the duration of the deadline that we imposed, which is long relative to the sub-second deadlines in some previous studies (for example, ref. 30), was particularly well-suited to revealing signatures of time-dependency. This prospect may point to a dependence of precisely-timed within-trial adjustments of decision policy on neural systems dedicated to the estimation of relatively long temporal intervals[55] or, perhaps complementarily, to constraints on the timescale over which the neural mechanisms responsible for these time-dependent adjustments operate. We note, however, that time-dependency operating over much faster timescales has previously been reported in the animal literature[21,22].

Although our behavioural and modelling results provide strong support for the existence of adaptive, time-dependent adjustments in subjects' decision policies under deadline, the

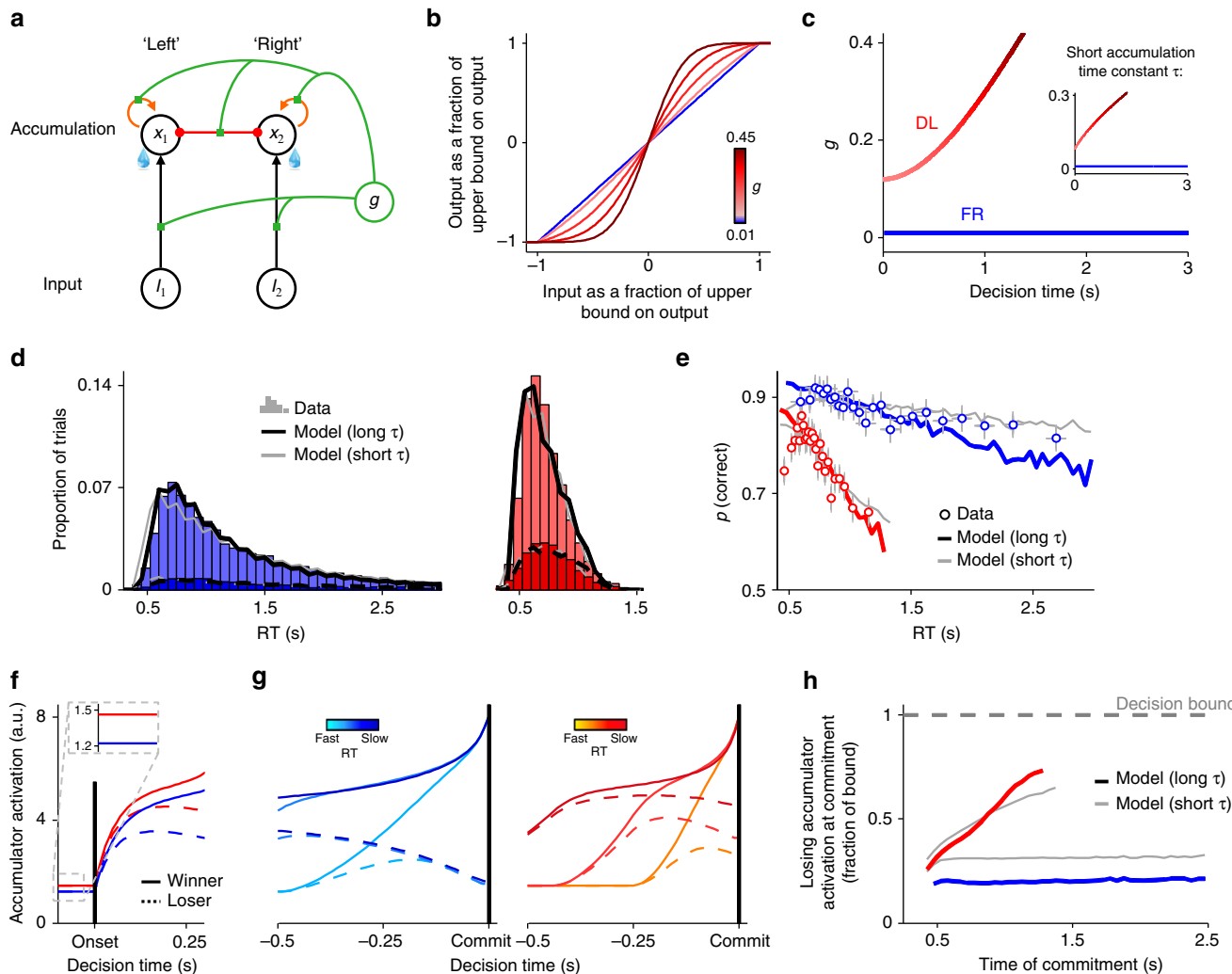

**Figure 6 | A simple network model generates signatures of speed pressure via global gain modulation alone.** (**a**) The model represented as a simple two-layer network in which choice-determining accumulation units are subject to recurrent excitation (orange), lateral inhibition (red) and leakage, and all connection strengths are modulated by global network gain (green). (**b**) The non-linear transfer function relating a unit's input to its corresponding output. The global gain parameter $g$ determines the slope of this function. (**c**) $g$ as a function of elapsed decision time in each speed regime. Informed by the pupil diameter results, there is a static offset in $g$ at the beginning of a DL trial coupled with a time-dependent increase in the DL but not FR regime. All other model parameters are fixed across regimes. Main plot shows $g$ time courses when the effective time constant of accumulation $\tau$ is unconstrained and takes a long value (555 ms); inset shows evolution of $g$ when $\tau$ is constrained to equal 167 ms (see ref. 52). (**d,e**) Observed and fitted RT distributions and conditional accuracy functions. Points in **e** depict mean ± s.e.m. accuracy of trials sorted by RT into 25 equal-sized bins. (**f,g**) Activation time-courses of the winning and losing accumulator units, simulated using fitted model parameters with unconstrained $\tau$ and aligned to motion onset (**f**) and decision commitment (**g**, sorted by RT into 3 bins). Global gain modulation qualitatively produces both a pre-motion offset in activation of both accumulators and a time-dependent increase in common activation of both accumulators under deadline. (**h**) Activation of the losing accumulator at decision commitment plotted as a function of commitment time for each speed regime.

time-course of these adjustments was not strictly optimal. If afforded high flexibility of form, the optimal, reward-maximizing policy given our task design is to adopt a predominantly static criterion on accumulated evidence that steeply declines to zero at a latency determined by the subject's level of deadline timing uncertainty[20,30]. In our data, however, the shape of the observed time-dependency approximated the reward-maximizing case only if the criterion change in these calculations was constrained to be linear in time. This approximately linear trajectory was strikingly preserved across all subjects, and is similar in form to the time-dependent policy adjustments that have been observed in brain and behaviour in non-human primates[21–23]. Collectively, these findings could point to basic limitations of the neural mechanisms responsible for generating time-dependency in the decision process and, consequently, to constraints on the

application of such policy adjustments for reward rate maximization in different settings.

Using lateralized 8–14 Hz oscillations in the EEG as a proxy for decision-related motor preparation[36–38], it was possible to establish that speed emphasis appeared to affect the dynamics of decision formation via a combination of static and time-dependent urgency, rather than a change in the level of the decision bound. Specifically, while the motor signals reflecting preparation for the ultimately chosen alternative reached a stereotyped pre-response level across speed regimes, we observed a deadline-induced bi-lateral increase in baseline preparation prior to decision onset, coupled with peri-decisional common activation of both effectors that was greater for slower responses. These effects have clear analogues in previous reports. In humans, functional MRI studies indicate that speed emphasis is at least

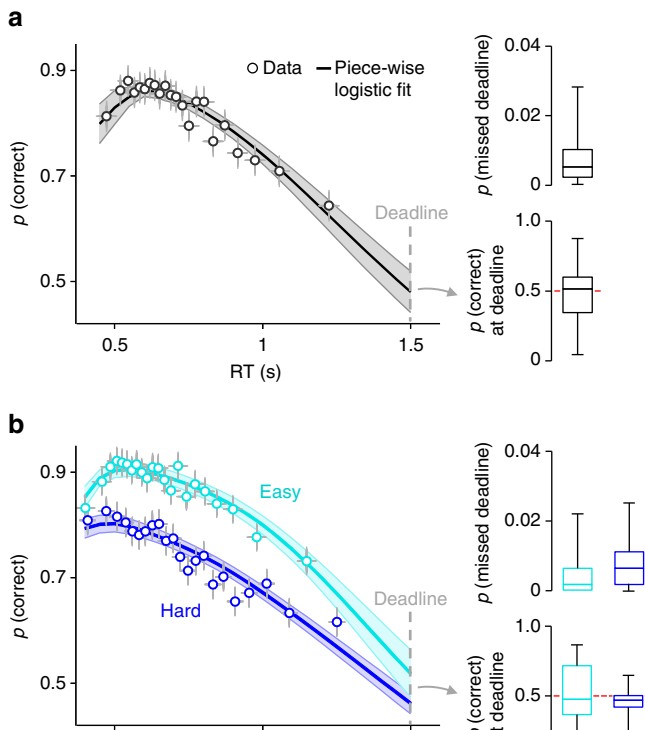

**Figure 7 | Behavioural signatures of time-dependency under mild speed pressure.** (**a**) Conditional accuracy function, proportion of missed deadlines and estimated accuracy at deadline in a context of mild deadline-induced speed pressure. Subjects performed the random dot motion task at a single level of discrimination difficulty. For the conditional accuracy functions, points indicate mean accuracy of trials sorted by RT into 22 equal-sized bins and line shows the mean of the best fits of piece-wise logistic regressions to each subject's single-trial data. Error bars and shaded areas indicate ± s.e.m. of data points and regression lines, respectively. Box plots show the sample median (centre line), interquartile range (box) and full range (whiskers). (**b**) Equivalent plots from a second study in which subjects performed under two levels of discrimination difficulty.

partly generated by an increase in the baseline activation of a network of decision-related brain regions (reviewed in ref. 1). In monkeys, speed pressure has similarly been shown to manifest in higher baseline firing rates of single neurons that reflect the developing decision process, but also in a time-dependent, evidence-independent increase in firing rates indicative of a growing urgency signal[21–24]. In light of the latter, our findings suggest that the mechanistic basis of SAT adjustment may be conserved across species. They also highlight that correlates of this mechanism in action are observable at the level of scalp electrophysiology.

How such urgency might be generated in the brain has been the subject of recent interest. Biophysically detailed computational analyses indicate that modulation of the gain of neural processing is one plausible mechanism for generating urgency in decision circuits[19,26,27,39,40]. Building on associations between pupil diameter and the activity of brainstem neuro-modulatory systems (the LC–NA system in particular[41–43]) and the established role of these systems in global gain modulation[44–47], we provided empirical support for these ideas. We observed that pupil diameter during the pre-motion period was reliably larger under deadline and that decision-related pupil dilation increased with elapsed time specifically in this condition, thereby identifying pupillometric counterparts to the static and

dynamic signatures of urgency that were observed in brain and behaviour.

Pupil-linked neuromodulatory nuclei like the LC project to almost the entire cerebral cortex and their associated neuromodulator release exerts a multiplicative influence on neural dynamics that interacts with the strength and location of ongoing processing[44,46,56]. The implication of such a general, global mechanism for urgency generation is appealing in part because it affords a simple yet very powerful means for affecting decision-making that is not specific to any one sensory input modality or effector. Indeed, global gain modulation might plausibly account for recent observations that urgency manifests not only in the firing rates of neurons that track the evolving decision process, but also in the gain of sensory inputs to decision circuits[57], in more downstream neurons involved directly in movement execution[23], and in the 'vigour' of task-irrelevant saccades during manual reaching decisions[28]. Similarly, such a global mechanism for urgency generation affords a parsimonious explanation for two potentially related observations in our data that can be viewed as distinct from effects of speed emphasis on the evidence accumulation process *per se*: the marginally quicker non-decision times under deadline; and, the deadline-induced desynchronization of baseline 8–14 Hz EEG power over occipital scalp, a phenomenon which itself has been associated with increased gain of responses in visual cortex[58].

We adapted a simple neural network model[12] to verify that global gain modulation alone can produce the behavioural and neural effects of deadline-induced speed pressure. To this end, a key qualitative effect that required reproduction was the time-dependent increase in the common activation of both accumulators under speed pressure, which we observed in EEG motor preparation signals and has also been reported in the firing rates of single neurons involved in decision formation[22,23]. Interestingly, this effect breaks the winner-take-all attractor dynamics characteristic of typical configurations of biophysically detailed spiking network models of decision-making[59,60], and in general cannot be generated via gain modulation alone in simpler models (like the DDM) that do not incorporate a recurrent excitation component. In our model, the effect was reproduced by constraining the recurrent excitation of accumulators to be stronger than the lateral inhibition between them, such that increasing network gain over time effectively heightened the dominance of excitation over inhibition and led to building activation in both accumulators. In principle, though, such an effect would also be produced by a network with more balanced excitation/inhibition in combination with stronger gain modulation for excitatory than inhibitory connections (cf. refs 39,40). Physiological data exploring potential differences in the neuromodulation of NMDA and GABAergic receptors could be highly informative about whether such dynamic changes in the ratio of excitation to inhibition occur in decision circuits.

Finally, although our analyses indicate that global gain modulation is sufficient to produce the qualitative effects of deadline-induced speed pressure, this does not preclude the existence of other sources of urgency in the brain. In particular, it has been suggested that enhanced speed pressure leads to the recruitment of cortico-basal ganglia pathways that in turn generate an effective additive input, via release from inhibition, to decision and motor circuits[1,61,62]. Such an influence could act in tandem with multiplicative gain modulation to amplify both the static and time-dependent effects of speed emphasis observed here.

## Methods

**Subjects.** We report data from four independent cohorts of subjects. All subjects were over the age of 18, had normal or corrected-to-normal vision, and no history

of psychiatric illness or head injury. They provided written informed consent and all procedures were approved by the ethics committee of the Leiden University Institute of Psychology. Subjects received either course credit or a fixed or performance-dependent gratuity for their participation. Sample sizes were pre-planned and consistent with other studies of human decision-making, from our lab and others, that interrogated similar physiological signals and invoked similar analytical methods. Cohort-specific information is given in the Supplementary Methods.

**General task procedures.** All reported studies employed variants of the random dot motion (RDM) paradigm[32]. Here we report general task procedures; additional study-specific information is provided in the Supplementary Methods.

Stimuli were presented using the Psychophysics Toolbox[63] for Matlab. Subjects maintained fixation on a centrally presented cross and decided whether the dominant direction of motion of a cloud of moving dots centred on fixation was either leftward or rightward. The difficulty of these discriminations was determined by the coherence $c'$ of the cloud of dots. Subjects indicated their decision by pressing one of two spatially compatible response keys with their left/right index fingers and were typically given feedback about the accuracy of their response on each trial. The interval between a subject's response and subsequent trial onset was randomly drawn from uniform distributions with study-specific bounds, and as such the response-to-stimulus interval did not depend on RT from the previous trial. Subjects completed initial practice and difficulty calibration routines prior to main testing, and in most cases were monetarily rewarded and punished in a performance-dependent manner under study-specific incentive schemes. Of particular note, subjects in studies 1 and 2 received $0.5\,¢$ for every correct decision, lost $0.5\,¢$ for every incorrect decision and, in the DL regime of these studies, lost $5\,¢$ if they failed to respond within a temporal deadline of 1.4 s following motion onset. This relatively heavy punishment for missed deadlines serves to heighten the deadline-induced speed pressure and was implemented to mitigate the 'accuracy bias'—a prioritization of accurate decisions even at the cost of decreased reward rate—that human subjects sometimes display on choice RT tasks[4,54] and could attenuate time-dependent adjustments to decision policy[30].

In all studies, subjects attended a single testing session and discrimination difficulties were calibrated to yield approximately similar response accuracies across individuals. In study 1, 21 subjects performed 8 blocks of 180 trials at a fixed discrimination difficulty per individual (equating to 75% accuracy under deadline), with 4 blocks under a deadline of 1.4 s and 4 under free response. In study 2, 23 subjects performed 10 blocks of 90 trials, split into the same DL/FR conditions at the same subject-specific difficulty setting. In study 3, 26 subjects performed 5 blocks of 100 trials, this time all at a lower discrimination difficulty (equating to 85% accuracy) and a response deadline of 1.5 s. In study 4, 21 subjects performed 8 blocks of 160 trials, again under a 1.5 s deadline but now two difficulty levels (corresponding to 70 and 85% accuracies) that were interleaved in random order across trials within each block. In all cases subjects were familiarized with the task and encouraged to form stable estimates of the precise timing of the response deadline during practice routines.

Several task design features were implemented to minimize contamination of EEG (study 1) and pupillometric (study 2) signals: Upon response execution, coherent dot motion transitioned to purely random motion for a fixed time to avoid sensory or feedback-related transients at the time of response execution and minimize post-decisional evidence accumulation[64]; a mask of static dots was displayed during the inter-motion interval to avoid luminance-related transients at motion onset; and, during pupillometry, the inter-trial interval was extended to negate contamination of the baseline period by the previous trial's dilation response, and post-response feedback was not provided so that the dilation response was not contaminated by feedback-related processes.

All statistical tests were two-tailed. In cases where data were non-normal (as determined by the Kolmogorov–Smirnov test), non-parametric tests were used as described below.

**Analysing empirical conditional accuracy functions.** Single-trial logistic regression was used to estimate mean accuracy as a function of RT (the CAF), for each task condition and subject. To account for both the dominant decreasing portion of the observed CAFs and an initial increasing portion due to a small percentage of inaccurate premature responses, we constructed an algorithm that minimizes the combined sum of squared errors of piece-wise logistic regressions of accuracy (1 = correct, 0 = error) onto RT, splitting trials before and after a temporal inflection point $\alpha$ such that

$$P_{correct} = \begin{cases} \left(1 + e^{-(\beta_0 + \beta_1 \times (RT - \alpha))}\right)^{-1}, & RT - \alpha \leq 0 \\ \left(1 + e^{-(\beta_0 + \beta_2 \times (RT - \alpha))}\right)^{-1}, & RT - \alpha > 0 \end{cases} \quad (1)$$

Here $\beta_0$ is accuracy at $\alpha$, $\beta_1$ is the slope of the CAF before $\alpha$, and $\beta_2$ is the slope of the CAF after $\alpha$. $\beta_1$ was constrained to be $\geq 0$ to reflect the fact that the left segment of the piece-wise fit should only account for the initial increasing portion of the CAF. This model was fit using Nelder-Mead simplex minimization to estimate the $\beta_0$, $\beta_1$ and $\beta_2$ parameters while conducting an exhaustive search of possible $\alpha$ values (step-size = 10 ms ending at 1 s). Whichever piece-wise segment is fit first determines $\beta_0$ and thus constrains the fit of the remaining segment;

therefore, the algorithm was run twice (left segment fit first and right segment fit first) for each $\alpha$ to find the true minimum[30]. In fits to FR trials from task 1, all RTs longer than 5 s were excluded.

To estimate accuracy at the time of the deadline (Fig. 1c), we used the piece-wise regression fits for each subject to calculate accuracy when RT = 1.4 s. The low number of trials immediately preceding the deadline prohibits a precise characterization of the shape of the empirical CAF at this time point. However, the above single-trial regression approach allows for a reasonable approximation by exploiting consistencies in the temporal evolution of the CAF. The appropriateness of this approach relies on any change in decision policy being gradual rather than abrupt, which appeared to be the case in our data given the smooth right tails of the DL RT distributions (Fig. 1b) and the shapes of the fitted urgency signals (Fig. 3b).

**EEG acquisition and analysis.** Continuous EEG was acquired from the first study cohort using an ActiveTwo system (BioSemi, The Netherlands) from 64 scalp electrodes, configured to the standard 10/20 setup and digitized at 512 Hz. Eye movements were recorded using two electrodes positioned above and below the left eye and two electrodes positioned at the outer canthus of each eye. EEG data were processed in Matlab via custom scripting and subroutines from the EEGLAB toolbox[65]. We describe the full EEG preprocessing pipeline in Supplementary Methods. In brief, we used Morlet wavelet convolution to estimate the power of effector-specific $\mu$ (8–14 Hz) oscillations, which we then employed as an index of decision-related motor preparation[36–38]. Pre-stimulus $\mu$ power was measured as the mean power from $-0.3$ to $-0.1$ s preceding motion onset. 8–14 Hz power in the human EEG is subject to a prominent decrease in the immediate post-stimulus period that is generated over lateral occipital scalp (Fig. 2b, middle inset) but spreads anteriorly and contaminates early portions of the motor preparation signals of interest here. For this reason, we restricted our analyses of post-onset $\mu$ signals to the period immediately preceding response execution, which is less susceptible to contamination by this early occipital response. Pre-response $\mu$ was measured as the mean power from $-0.17$ to $-0.05$ s preceding response execution (a window that was centred on the latency of peak desynchronization in the response-aligned grand-averages and chosen in a manner that was orthogonal to potential RT and condition × RT effects; Fig. 2b, right).

We interrogated relationships between $\mu$ power and decision-making behaviour via a series of single-trial within-subjects regression models that are described in the Results section and specified in full in Supplementary Methods. For all analyses, FR trials with RT > 5 s were not included. Additionally, to mitigate the influence of the stereotyped stimulus-evoked occipital response (Fig. 2b, middle inset) on the motor $\mu$ signals of interest here and also exclude 'fast guesses' from analysis, we discarded trials with RT < 0.5 s from all EEG analyses. For all models, the group-level significance of effects represented by individual regression coefficients ($\beta_i$) was tested via one-sample $t$-test ($H_0$: $\beta_i = 0$).

In one analysis, we examined $\mu$ signals for evidence of a time-dependent influence on motor preparation that varied with speed pressure. With the decision bound fixed, the difference in activation between accumulators at the time of decision commitment can provide a proxy for the strength of an additive urgency signal. Specifically, if the decision process is driven by evidence accumulation without urgency in a winner-take-all competitive network (for example, ref. 59), then the winning accumulator will inhibit the losing accumulator and the difference in their activations will be large by the time the decision bound is reached. On the other hand, if both accumulators also receive additional, evidence-independent input due to urgency, then the common activation of both accumulators at the time of commitment should increase in proportion to the strength of the urgency signal at that time and there will be less of a difference between accumulators when urgency is stronger[22,23]. Thus, the shape of the urgency signal over time can be approximated by examining, across all levels of RT, either the raw amplitude of the losing accumulator at the time of commitment, or the difference in activation between accumulators at that time. We focus on the latter because a difference metric is, in principle, more robust to any RT-dependent contamination of $\mu$ signals by the strong bi-lateral occipital response described above (Fig. 2b, middle inset). However, we also examined the pre-response amplitude of the ipsi-lateral $\mu$ signal alone for time-dependency, and this analysis yielded similar effects (Supplementary Fig. 6).

We also explored the relationship between decision-making behaviour and effector-specific power in the $\beta$ frequency band (14–30 Hz), which has also been linked to decision-making[36–38]. However, $\beta$ power did not exhibit several of the critical effects that we identified in the $\mu$ signals (Supplementary Fig. 7).

**Pupillometric acquisition and analysis.** Pupil diameter and gaze position of the second study cohort were recorded at a sampling rate of 250 Hz using an Eyelink 1000 eye-tracker (SR Research, Canada), and analysed in Matlab. After data cleaning and artifact rejection (see Supplementary Methods), we employed a paired-samples $t$-test and single-trial within-subjects regressions to examine speed regime effects on pre-motion pupil diameter and post-onset pupil dilation, respectively. Pre-motion pupil diameter was measured as the mean, unbaselined pupil diameter from $-0.2$ to $0$ s relative to motion onset. Evoked pupil dilation was measured as the mean pupil diameter within a 0.7 s window centred on the latency of peak dilation in the response-aligned grand-average waveforms from each speed regime (0.55 s post-response in FR, 0.75 s post-response in DL; Fig. 4b),

baselined relative to the pre-stimulus interval, though we also show that the reported effects are robust to different measurement windows (Supplementary Fig. 4).

The phasic input to the peripheral system controlling pupil diameter was modelled as a linear combination of three temporal components: transients at motion onset and response, and a sustained component throughout the intervening period[49,50]. For each subject and speed emphasis condition, eight different models were constructed in which the sustained component took one of the following shapes (Fig. 5a): (1) a boxcar with constant amplitude throughout the decision interval; (2) a linear up-ramp that grew in amplitude with increasing decision time; (3) a ramp-to-threshold; (4) a linear decay with a starting amplitude that was larger for slower RTs but whose amplitude always terminated at zero; (5) a linear decay-to-threshold which began at a fixed amplitude and terminated at zero; and, (6–8) versions of the boxcar, up-ramp and down-ramp in which the sustained component for each trial was normalized by the number of samples in that trial's decision interval, thereby negatively modulating these components by RT. To fit each model, a vector of concatenated pupil dilation waveforms, from 0.2 s pre-stimulus to 2.5 s post-response, was regressed onto a general linear model composed of the three temporal components (onset, sustained, response) convolved with a pupil impulse response function[51]:

$$h(t) = t^w \times e^{-t(w/t_{max})} \qquad (2)$$

where $w = 10.1$ and $t_{max} = 930$ ms (matching the function used in refs 49–51, though the key findings were robust to specific parameter combinations; Supplementary Fig. 5). Model fit was assessed using the Bayes Information Criterion (BIC) for models estimated via least squares:

$$BIC = n + n \log(2\pi) + \log(SSR/n) + (k+1)\log(n) \qquad (3)$$

where $n$ is the number of samples, SSR is the residual sum of squares, and $k$ is the number of free parameters. The relative goodness of fit between two given models was assessed non-parametrically by subjecting difference values ($BIC_1$-$BIC_2$) to Wilcoxon signed rank tests.

**Drift diffusion modelling.** Behavioural data from the study 1 cohort were fit with several versions of the DDM for two-alternative decisions[10], both with and without an urgency component. In its most basic form, the DDM assumes that noisy sensory evidence is accumulated from a starting point $z$ at drift rate $v$ and a decision is made when a criterial amount of cumulative evidence reaches one of two opposing boundaries corresponding to either choice option. The distance between boundaries is the boundary separation $a$, while the model ascribes all non-decision-related processing to a non-decision time parameter $t_{er}$. Noise in the evidence is determined by $s$, the s.d. of a zero-mean Gaussian distribution, and is fixed at 0.1 to scale all other parameters[66]. Given any combination of the above parameters, the DDM yields a flat CAF and thus cannot account for the negative CAF slopes that we observed in our data. However, including between-trial variability in drift rate (normally distributed with s.d. = $\eta$) allows the model to produce decreasing CAFs[33,34], and so we also included this parameter in our model fits. In all models, $z$ was fixed at $a/2$. Thus, what we refer to as the 'standard DDM' had, at a minimum, four free parameters ($v$, $\eta$, $a$, $t_{er}$).

Informed by our EEG findings, we also considered DDM variants that incorporate an additive urgency component. In the 'urgency DDM', decisions are determined by the states of two perfectly anti-correlated accumulators that are subject to regular drift diffusion, and are each summed with the same time-varying, evidence-independent quantity (the urgency signal). A decision is made when the total activation (diffusion + urgency) of one of the accumulators passes a common decision bound (fixed at 1 for all conditions and subjects). The shape of the urgency signal was parameterized by a logistic function:

$$u(t) = u_0 + \left(1 - e^{-(t/\lambda)^k}\right) \qquad (4)$$

where $u(t)$ is the magnitude of the urgency at decision time $t$, $u_0$ is the static component of the urgency (that is, the value of $u$ when $t = 0$), and $k$ and $\lambda$ are shape and scale parameters that determine the shape of the time-dependent component of the urgency. The logistic function was chosen because it can produce a variety of different shapes of urgency signal (concave, convex, approximately linear, flat) using few free parameters. Although conceptually distinct, this urgency model is mathematically identical to a model in which the standard DDM is coupled with time-varying decision bounds. The urgency DDM had a minimum of five free parameters ($v$, $t_{er}$, $u_0$, $k$, $\lambda$), or six in cases where $\eta$ was also included.

We fit a number of models with varying parameter constraints (Supplementary Table 1) and estimated parameters for each model and subject using maximum likelihood estimation procedures that are described in the Supplementary Methods. Of particular note, for the urgency DDM we invoked a method for analytically deriving first passage time densities through continuously differentiable time-varying bounds[67]. This approach is based on the analysis of renewal equations and described in detail by Smith[68] and Zhang et al.[69]. The Supplementary Methods also contain a detailed description of our approach for estimating the optimal, reward-maximizing time-dependent urgency signals, given our task, for a representative set of time-invariant parameters.

**Leaky competing accumulator modelling.** To interrogate effects of global gain modulation on decision-making, a modelling approach must be employed that allows basic features of neural information processing, such as the relative strength of recurrent excitation and lateral inhibition, to be dissociated; these properties of a neural network, which are not distinguished in the more abstract DDM, determine the nature of effects of gain modulation on accumulation dynamics and decision-making behaviour[39,40]. We therefore modelled gain modulation by adapting the LCA model[12], which is built upon such principles of neural computation and offers a tractable means of interrogating gain effects without the level of complexity inherent in more biophysically detailed models of decision-related neural population dynamics[39,40,59]. Note that we did not employ this model for earlier quantitative model comparison because, despite its simplicity relative to more biophysically plausible neural networks, it is under-constrained (see Supplementary Methods).

In the two-alternative LCA model, decision-making is driven by a simple two-layer neural network consisting of two units over which external input is represented, and two accumulator units, one for each response alternative, that determine choice (Fig. 6a). Each unit, which represents a population of functionally equivalent neurons, is characterized by two variables: its activation, which captures the net input to the unit, and its output, which is related to activation via a nonlinear transfer function (see below). The activation values of the first (correct) and second (incorrect) input units are $I_1$ and $I_2$, respectively, and their associated outputs to the accumulator units are $f(I_1)$ and $f(I_2)$. The momentary change in the activation of each accumulator unit $x_i$ can be approximated by the following finite difference equations[12]:

$$\begin{aligned} \Delta x_1 &= f(I_1) - \lambda x_1 + \alpha f(x_1) - \beta f(x_2) + f(N(0, \sigma)) \\ \Delta x_2 &= f(I_2) - \lambda x_2 + \alpha f(x_2) - \beta f(x_1) + f(N(0, \sigma)) \end{aligned} \qquad (5)$$

and the accumulator units are subject to a lower bound on activation such that:

$$\begin{aligned} x_1(t+1) &= \max(0, x_1(t) + \Delta x_1) \\ x_2(t+1) &= \max(0, x_2(t) + \Delta x_2) \end{aligned} \qquad (6)$$

In Equation (5), $\lambda$ represents the leak or decay of activation over time, $\alpha$ represents recurrent excitation, $\beta$ represents lateral inhibition, and $N(0,\sigma)$ is a zero-mean Gaussian-distributed noise term with s.d. = $\sigma$. A decision is made in the model when the activation of one of the accumulator units exceeds a decision bound $A$.

Note that, with the exception of the leak, every term that contributes to $\Delta x_i$ in Equation (5) is passed through the transfer function relating a unit's activation to its output. Varying the slope of this function provides a natural way to implement global gain modulation in the LCA. In accordance with extensive previous modelling work (for example, refs 27,45,47,70,71), we assumed that the transfer function is sigmoidal in shape. The sigmoid places upper and lower bounds on output and thus prevents runaway activation in cases where the effective recurrent excitation is greater than the leakage (i.e. $\alpha f(x_i) - \beta f(x_{i' \neq i}) > \lambda x_i$), which can happen when gain is high. We favored a transfer function that becomes linear within the range of possible outputs as gain approaches 0, thus approximating the threshold-linear function employed in the original LCA model[12], and step-like as gain approaches $\infty$. This function took the following form[70]:

$$f\left(x \mid 0, \frac{1}{g}\right) = \begin{cases} -\theta, & x \leq -\theta \\ -\theta + 2\theta \frac{\int_{-\theta}^{x} \varphi\left(y \mid 0, \frac{1}{g}\right) dy}{\int_{-\theta}^{\theta} \varphi\left(y \mid 0, \frac{1}{g}\right) dy}, & -\theta \leq x \leq \theta \\ \theta, & x > \theta \end{cases} \qquad (7)$$

where $\varphi(0, 1/g)$ is the cumulative function of a normal distribution with mean = 0 and s.d. = $1/g$, and $\theta$ determines the symmetric upper and lower bounds on output. The gain parameter $g$ determines the steepness of the non-linearity in the function (Fig. 6b).

Informed by our pupillometric findings, we realized urgency through global gain modulation in this adapted LCA model by allowing both the baseline offset and within-trial time-varying trajectory of the $g$ parameter to vary with speed regime (Fig. 6c). All other model parameters were fixed across speed regimes. Full specifications of the remaining model parameters, fitting procedures, and approach used for simulating accumulator time-series are all provided in the Supplementary Methods.

**Data availability.** The data and computer code that support the findings of this study are available from the corresponding author on request.

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

## Acknowledgements

The authors thank Redmond O'Connell and Konstantinos Tsetsos for helpful discussions. This work was supported by an ERC Consolidator Grant awarded to S.N.

## Author contributions

P.R.M. and S.N. conceived and designed the experiments. E.B. and P.R.M. collected the data. P.R.M. analysed the data and fit the models. P.R.M. and S.N. wrote the manuscript and all authors approved the final version.

## Additional information

**Competing financial interests:** The authors declare no competing financial interests.

**DOI: 10.1038/ncomms14299**    **OPEN**

# Erratum: Global gain modulation generates time-dependent urgency during perceptual choice in humans

Peter R. Murphy, Evert Boonstra & Sander Nieuwenhuis

*Nature Communications* 7:13526 doi: 10.1038/ncomms13526 (2016); Published 24 Nov 2016; Updated 18 Jan 2017

In Fig. 4c of this Article, the colour bars were inadvertently changed from graded to solid during the production process. The correct version of Fig. 4c appears below as Fig. 1.

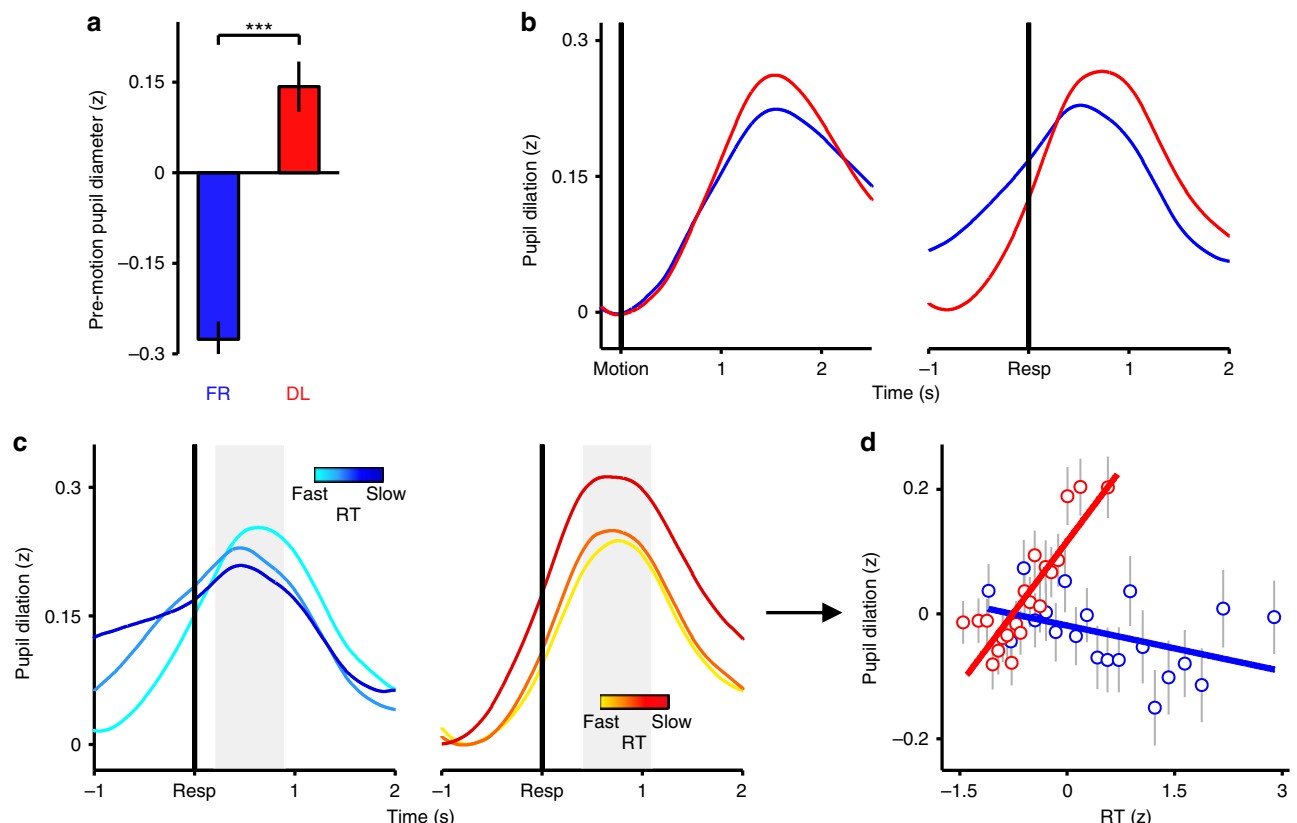

**Figure 1**

