## [Peer Review File · Nature Communications]

Reviewers' comments:

Reviewer #1 (Remarks to the Author):

This paper investigates a fundamental question in decision science, namely if human participants can modulate decision thresholds dynamically over the course of a trial (in a time adaptive fashion). Researchers present convergent evidence from behavioral, EEG and pupillometry that shows that humans can modulate their decision thresholds in a time adaptive fashion such that they integrate evidence while making sure that a decision is made prior to the deadline (which ensures 50% accuracy [irrespective of penalty] and avoids penalty in the particular experiments conducted in the current study).

The work is certainly interesting and relevant. Experiments were conducted well and the different approaches researchers adopted in their experiments are indeed complementary. Other research groups have been working on decision-making under response deadlines for several years and could not observe time-adaptive threshold modulation (at least not to the extent that has been observed in the current work). At a point where it seemed that humans cannot dynamically collapse decision thresholds, coming across the findings of the current study was very timely and pleasing.

Other researchers have tried penalty conditions (e.g., -1 pts for being late and +1 for correct responses), utilized dynamic auditory urgency signals, tried different levels of motion coherences, etc. None of these experiments showed the convergence of participants' micro speed-accuracy tradeoff curves to chance level at the deadline, and the observed amount of decline in accuracy could be explained by other factors such as increased drift-rate variability.

Thus, it is great that the authors could get their participants achieve the reported level of performance; however this makes me wonder what procedural differences would have led to differential results between the current paper and these other studies. One possibility is that the deadline used in the current work is a supra second interval whereas in the other work, they were mostly sub-second intervals (~800 ms). Keeping this in mind, is it possible that the time-adaptive threshold modulation can be achieved only over time scales that is the subject to perceptual timing -underlain by corticostriatothalamic loop- rather than - cerebellar- motor timing? This is a relevant issue to address in the discussion. Previous work that authors cited have also discussed the possible relationship between interval timing and time-adaptive threshold crossing. On a related note, it is not very clear from the main text or SOM what would have led to the relatively long free-response RTs gathered in the current study. One possibility is the higher limit for premature responses. I think overall more information should be provided (preferably in the main text) regarding the key procedural details (e.g., average coherence, etc.).

See below for a more major issue:

There is a large difference between the reward attained for correct responses and penalty amount associated with missing the deadline. Frazier & Yu (2007) and Karsilar et al (2014)

have formulated the optimal threshold collapse trajectories (as a function of trial time) for those cases in which there was no explicit penalty for missing the deadline. It is these conditions for which the optimal strategy is minimal threshold setting (i.e., meeting thresholds) at the deadline.

When there is a large relative penalty associated with missing the deadline (e.g., -10:1) and given endogenous timing uncertainty regarding the deadline, reward-rate maximization requires minimal threshold settings (e.g., threshold = $x(0)$) much earlier than the deadline. A statistical decision theoretic approach to the decision scenario at hand would reveal this relation. Consequently, under this account accuracy should converge on the chance level performance prior to the deadline. However, this is not what has been observed in the current work. Thus, when evaluated within the framework of optimality, it is not clear how the findings of the current study are different from the results of Karsilar et al (2014) - note that optimal threshold collapse trajectories should be different between these two studies given the payoffs. I think addressing this issue would strengthen the manuscript and contribute to its scope.

There is not enough data at the critical region right before the deadline. One way to address this problem would have been shortening the deadline (although this has not paid off in other studies) or increase the number of data points.

It is not clear if the RT distributions were based on pooled or vincentized data.

Reviewer #2 (Remarks to the Author):

This manuscript describes several experiments on human perceptual choice (all using the random-dot motion discrimination task), that examine the putative influence of a time-dependent "urgency" signal. In experiment 1, subjects are tested in a Free Response (FR) condition as well as a Deadline (DL) condition, in which responses longer than 1.4s incur a heavy penalty (equal to -10x the value of a single correct choice). Behavior is analyzed using conditional accuracy functions (CAFs) showing a strong effect of urgency in the DL condition, consistent with an approximately linear urgency signal. EEG data in the 8-14Hz band shows: 1) reduced power in the DL vs. FR condition at baseline and at motion onset; 2) condition-independent power contralateral to the responding hand just prior to response onset; 3) an RT-dependent difference in the contrast between contra-vs-ipsi power in the DL condition. The authors suggest this reveals the presence of a time-dependent signal (especially in the DL condition) that is effector-independent, consistent with an urgency signal. In Exp 2, subjects perform almost the same task, but adjusted to allow measurement of pupillary responses. The data shows: 1) a significant increase in the baseline pupil diameter in the DL vs FR task; 2) a time-dependence on pupil size that is consistent with a linear ramp; 3) an RT-dependence of pupil size at response time in the DL condition. Both of these sets of results tell the same story: That in the FR task there is a flat urgency signal, but that signal becomes very steep in the DL task. Two kinds of computational models are applied to this data: a classic drift-diffusion model (DDM) with or

without additive urgency, and a leaky competing accumulator (LCA) model with a time-dependent transfer function. Simulations show that a DDM without urgency cannot capture the behavioral data, but a DDM with urgency can, and the LCA with urgency can capture both the behavioral data as well as neural processes inferred from EEG and pupillary analyses. The take home message is that there exists a time-dependent urgency signal that modulates the gain of processing, especially when there is time pressure. Finally, the authors show that this generalizes to conditions with only mild time pressure. They re-analyze data from a previous study (Exp 3) and a new experiment (Exp 4) in which there was no heavy penalty for missed deadlines. Again, they use conditional accuracy functions to show a strong effect of urgency on behavior.

Overall, I found this to be a very interesting manuscript presenting highly compelling data. Most impressive was the good qualitative match between urgency signals (in Exps 1 & 2) derived in three different ways: from behavior, from EEG, and from pupil diameter. This was strengthened by solid and thorough modeling, using an abstract formalism (DDM variants) and using a neural circuit model (LCA). In all cases the conclusions were congruent. Finally, the additional experiments (3 & 4) further show the generality of the conclusions to other situations. This provides a strong counter argument to recent papers claiming that urgency does not play a role in human perceptual choice.

Although I liked this paper very much, I have some suggestions for further improvement.

GENERAL COMMENTS:

1. As the authors are aware, a dropping conditional accuracy function (CAF) can be produced without any actual rising urgency. This is stated explicitly on line 548, in the methods, but it is not clearly acknowledged in the main body of the paper. This could create the impression for some readers that if the CAF is decreasing, then an urgency signal must be present. I think it's better to clarify, up-front, that a dropping CAF is expected in any paradigm in which subjects are allowed to determine their own response time - because attention/arousal fluctuates, trials in which it happened to be low will tend to have long RTs and also be less accurate, skewing the distribution to create a dropping CAF. The authors already allow for this effect by permitting between-trial variability in gain (in both DDM and LCA models), and their simulations show that this is enough to generate the mildly decreasing CAF in the FR condition but not the dramatically steeper one in the DL condition (Fig. 3h). Nevertheless, I think their data might make it possible to control for this confounding effect even more convincingly. In particular, in study 2, can't one use the baseline pupil diameter as an estimate of the attention/arousal on a given trial? If so, then one could select out trials with a similar baseline pupil diameter and then repeat the analyses in Fig. 1. Any remaining negative slope of the CAF could then be more confidently attributed to urgency. A simulation of a DDM with fixed bound and no variability in gain (drift rate) would presumably not show a negative CAF in either condition.

2. Although the authors strongly argue against one of the assumptions of the DDM, the fixed bound, they still adhere to the assumption of perfect integration, at least in the text and DDM simulations. However, it has recently been argued that perfect integration does

not occur, even in the random-dot motion discrimination task (Carland et al. 2016, J Neurophysiol), but that instead, the motion signal is processed with a low-pass filter with a time constant of less than 250ms. This translates to a large leak term in a model such as the LCA used here. As noted by Ditterich (2006), once one allows for time-dependent gain (urgency), data from motion tasks can be explained without perfect integration. This implies that the authors could also relax that assumption and simulate all of their data using models with a short time constant. The LCA model used here is in fact a "leak dominant" LCA that has a time constant determined by: $\text{timestep} / (\lambda - \alpha - \beta)$, or $0.01 / (0.201 - 0.110 - 0.073) = 555\text{ms}$ (assuming I got the math right). That is not compatible with the data of Carland et al. These parameter values were found by assuming a constant urgency $g=0.01$ for the FR task. However, it would be interesting to see whether you can still fit the data from your FR and DL tasks if instead of assuming a constant g for the FR task, you constrain the time constant to be shorter than 250ms and let the other parameters vary, including the urgency signal. I understand that in these fitting exercises one does not want to let all parameters vary, and you've chosen to fix g for the FR task on the basis of your data. However, unless you claim that there is something different about your task and the task of Carland et al., you can instead place a constraint on the time constant and let g vary over time (This might produce a non-constant urgency in the FR task, but it might not: The strongly exponential right-tail of your RT distributions might in fact be explainable even if the time constant is short and urgency is flat - it would be something akin to a low-pass filter motion detector with a "survival function"). It would be worth trying, because with the same model you could probably also simulate all of the data from the Carland et al. study, and abandon the assumption of perfect integration.

SPECIFIC COMMENTS:

Lines 77-87: I think more information should be provided in the main text on some of the details of the experiment. In particular, the payoff for correct choices (0.5c), the loss for errors (-0.5c) and the heavy penalty for missed deadlines (-5.0c) should be made clear. It would also be useful to explicitly state that the inter-trial interval was random within a range of 1.65-2.15s but that trials were not initiated on a rhythm (i.e. the timing of the next trial dependent at least in part on the RT in the previous trial). This is important because it motivates strategies for reward rate maximization.

Line 140: Desynchronization is misspelled.

Line 171 and elsewhere: The term "competition" is used here to mean common activation in two accumulators. But many people use that term to imply a direct process of competition, such as mutual inhibition through recurrent inhibitory connections. Saying "stronger competition" has very different implications given these usages, which can be potentially very confusing. I would recommend using a different term. For example, here you could just explicitly say "common activation ... can provide a proxy". The term competition is used elsewhere as well, and you might want to consider changing it. Alternatively, you could use the term, here and elsewhere, as long as you define it more explicitly and distinguish it from a mutual inhibition process.

Line 190: The urgency functions were fitted with a logistic function. But it seems to me that you'd get almost as good a fit with a purely linear function, for both FR and DL conditions. Is there any real advantage of adding this extra complexity? Note that the initial rising CAF and inflection can be produced with just large noise that causes early guesses.

Line 267: I don't understand the meaning of the subscripts Onset-PM and Resp-PM.

Line 281: I assume that when you say "first cohort" you mean Exp 1. This should be explicit.

Line 284: What does the word "latter" refer to?

Line 298: Not penalized in any way: Does this mean there were no monetary rewards or losses in Exp 3 and 4? This should be explicitly stated.

Line 315: What is meant by "cost of deliberation"? I think a more natural way of motivating a time-dependent urgency is to just assume that subjects want to maximize reward rate. That implies that time is in the denominator, thus automatically becoming a "cost". I'm not sure we need to add any other assumptions of deliberation costs.

Lines 424-426: If you don't put the numbers on payoffs and penalties in lines 77-87, then they should at least be here.

Lines 427-428: Did subjects do all of these trials in a single daily session or were they spread over two or more days?

Line 432: Were the easy and difficult trials fully interleaved or run in blocks?

Line 605: You can cite Standage et al. (2011) as implementing time-dependent gain in this way.

Fig. 2: I suggest adding labels for "FR" and "DL" to panels C and D, so that it's clear at a glance. I would also suggest adding a y-scale to the DL data in panel D.

Fig. 4: Why are the analysis windows in panel C different for FR and DL? Maybe there's a reason, but otherwise it looks like cherry-picking. My guess is you could just use the same window for both.

Fig. 3 vs. Fig 6: In Figure 3b you show a DL urgency function with a decreasing slope, and in Fig. 6c it has an increasing slope. But why the complexity? I would expect that a purely linear function would get you a fit that is almost as good, in both cases.

Supplemental materials: As noted above, I think many of the details of the experimental paradigms should be moved to the methods section in the main text. I found that I was confused about several important details, which could impact on interpretation, until I read the supplemental section.

Reviewer #3 (Remarks to the Author):

The manuscript investigates the mechanisms underlying human decisions based on perceptual evidence. The authors focus on the question if the employed decision policy depends on time or not, and combine behavioral, EEG and pupilometric evidence with sophisticated modeling to show that it does, and that mechanisms modulating the gain on evidence integration are able to mimic the presented observations. The work thus presents an interesting and timely contribution to the discussion on the exact mechanisms underlying human decisions, and in particular acts as an important counterargument to a series of recent papers claiming that the human decision policy is time-independent. The time-dependence of such a policy has already been established for non-human primates, but the concise set of provided evidence showing this for human decision-makers is, to my knowledge, novel.

My main concern about the current version of the manuscript is the over-interpretation of the EEG results. While EEG recordings give us some idea about the time-course and coarse localization of neural activity, using them to distinguish between collapsing bounds and an urgency signal seems far-fetched. There might be some non-linear mapping between neural activity and the EEG signals that produces similar measures if the system implements collapsing bounds. This is not to say the evidence does not support an urgency signal. I just think that the statements in the manuscript should be phrased more carefully. The same applies to the pupilometry measures, but in this case the authors are more careful in their statements ("plausible hypothesis").

A further concern about the EEG analysis is that the arguments in that section of the main text seem to be based on a model that is not elaborated on in the main text but rather only in Methods (and even there I don't find it particularly clear). In order to be able to follow these arguments, the authors should elaborate on the underlying model in the main text, maybe with a supporting figure panel?

Turning to interpretation of the pupilometry data, the authors suggest that "global gain" alone reproduces the effect of speed pressure on behavior and neural dynamics. In this context I wish that the authors could be more explicit in what they mean with "global gain", which, by itself, could mean anything. Surely, they don't mean that "gain" is cranked up everywhere in the brain, causing all the observed effects (and many more). In their model the "gain" is, in fact, fairly local. Thus, while "global gain" might sound appealing, I think that it is too unspecific to make falsifiable predictions, and so should be revised.

When analyzing behavior the authors seem to suggest two models, only one of which is supported by the data (Fig. 1c). However, the description of the models in the main text (around l91) is not very clear. Is the first model supposed to behave like in the FR condition up to the deadline, at which point accumulation is interrupted and a decision is cast? If this

is the case, the authors should also consider what happens if the deadline estimate is noisy (as will be the case) and conservative (biased away from the actual deadline). What would be the predictions of sharp cut-offs at noisy deadline estimates, and how would it compare to a model with a collapsing boundary? This should also be discussed in the context of the EEG / pupilometry analysis.

Other points that would require additional justification are some of the choices made in the analysis of the various signals. For example, why was the pre-response μ measured as the mean power from -170ms to -50ms (I474)? This seems a fairly specific, but on the other hand rather arbitrary choice. Are the results robust to changes in these values? Discarding trials with $RT < 0.5s$ for EEG analyses (I482) also seems restrictive, particularly for the DL condition. How was the pupil response function time constant (I532) chosen? ANY of those and similar choices warrant justification.

Minor comments:

I66: "recent empirical and model comparison reports have reinforced this stance" - another recent one to cite would be Voskuilen et al. (2016): "Comparing fixed and collapsing boundary versions of the diffusion model"

I69: "combination of static and time-dependent urgency" - it is unclear at this point what static and time-dependent mean. The same applies to I121, I138, etc.

I70: "urgency to optimally adapt to deadline-induced speed pressure" - optimality is never established.

I99: "negative-going" (and later occurrences) - isn't there a better way to express this?

I103: Fig. 1c and Methods describe a piecewise linear fit to the CAFs, but the main text only reports a single slope. If results in the main text come from standard logistic regression over the full RT range, then these fits should also be shown in Fig. 1c. The same applies to later uses of piece-wise linear fits.

I135: "desynchronization" - unexplained μ wave jargon that should be avoided to make the manuscript accessible for a larger audience. Why not just call it a drop in μ power throughout the manuscript (on the side/in general: such a drop can be caused by other effects than desynchronization, such that calling it such might be misleading).

I533: which definition of BIC was used here? The one shown on I533 uses different terms than the one I am used to (or that is, for example, on Wikipedia).

SI I272: was the $\log f(RT_i | \theta)$ split into correct/incorrect choices, or did you only model the reaction times?

We greatly appreciate the opportunity to re-submit our paper to *Nature Communications* and thank the reviewers and editor for taking the time to read our manuscript. We were very impressed by the quality and content of each of the reviews and feel that the manuscript has been strengthened significantly as a result of this review process. Please find below our responses to all comments, and a description of revisions made to the manuscript in light of these. We have submitted two versions of the revised manuscript: one with regular formatting and another with track changes turned on. All page numbers refer to the former.

Reviewer #1 (Remarks to the Author): *This paper investigates a fundamental question in decision science, namely if human participants can modulate decision thresholds dynamically over the course of a trial (in a time adaptive fashion). Researchers present convergent evidence from behavioral, EEG and pupillometry that shows that humans can modulate their decision thresholds in a time adaptive fashion such that they integrate evidence while making sure that a decision is made prior to the deadline (which ensures 50% accuracy [irrespective of penalty] and avoids penalty in the particular experiments conducted in the current study).*

The work is certainly interesting and relevant. Experiments were conducted well and the different approaches researchers adopted in their experiments are indeed complementary. Other research groups have been working on decision-making under response deadlines for several years and could not observe time-adaptive threshold modulation (at least not to the extent that has been observed in the current work). At a point where it seemed that humans cannot dynamically collapse decision thresholds, coming across the findings of the current study was very timely and pleasing.

Other researchers have tried penalty conditions (e.g., -1 pts for being late and +1 for correct responses), utilized dynamic auditory urgency signals, tried different levels of motion coherences, etc. None of these experiments showed the convergence of participants' micro speed-accuracy tradeoff curves to chance level at the deadline, and the observed amount of decline in accuracy could be explained by other factors such as increased drift-rate variability.

Thus, it is great that the authors could get their participants achieve the reported level of performance; however this makes me wonder what procedural differences would have led to differential results between the current paper and these other studies. One possibility is that the deadline used in the current work is a supra second interval whereas in the other work, they were mostly sub-second intervals (~800 ms). Keeping this in mind, is it possible that the time-adaptive threshold modulation can be achieved only over time scales that is the subject to perceptual timing -underlain by corticostriatothalamic loop- rather than -cerebellar- motor timing? This is a relevant issue to address in the discussion. Previous work that authors cited have also discussed the possible relationship between interval timing and time-adaptive threshold crossing.

We made three experimental design choices that differed from previous studies of deadline-induced speed pressure and, we suspect, likely combined to produce the strong time-dependent urgency effects that we report:

(1) In tasks 1 and 2, we administered a monetary punishment for missed deadlines that, to our knowledge, is more severe than any employed in previous work. The specific intention here was to mitigate the 'accuracy bias' that human subjects have been reported to exhibit without extensive training on choice RT tasks – that is, a prioritization of accurate decisions even at the cost of decreased reward rate (Balci et al., 2011, *Atten. Percept. Psychophys.*; Simen et al., 2009, *JEP: HPP*) – that could serve to strongly attenuate any adaptive time-dependent lowering to the decision bound.

(2) We calibrated the difficulty of the perceptual discriminations on our tasks to yield comfortably below-ceiling performance for all subjects. By contrast, at least one previous study of deadline-induced speed pressure (Karsilar et al., 2014, *Front. Dec. Neurosci.*) employed only relatively easy discriminations that were not calibrated to subject-specific discrimination thresholds. As a result, the average response accuracy in that study, even under a very stringent deadline of 800ms, was greater (mean = 90%) than the average accuracy in our free response condition (mean = 87%), and several subjects were at ceiling. The

ease of the task in this case, coupled with a lack of explicit penalty for missed deadlines, may have disincentivized subjects in this study from expending the necessary effort to implement strong within-trial bound adjustment, since their reward rates were already very high.

(3) As the reviewer points out, a third difference is that the deadline of 1.4s that we employed is substantially longer than in most previous work. We opted for this extended deadline because this previous work with shorter deadlines failed to report evidence for time-dependency, and also because we preferred broader RT distributions (and hence more variance to work with) for our electrophysiological and pupillometric analyses. Our behavioural piloting suggested that, coupled with our relatively difficult coherence levels, a deadline of 1.4s was appropriate. This design choice may have been an important factor for two related reasons. First, as the reviewer mentions, it has been suggested that estimation of supra-second time intervals relies on a distinct, so-called 'cognitive' neural system compared to the putatively 'automatic' system recruited for estimation of sub-second intervals (e.g. Lewis & Miall, 2003, *Curr. Opin. Neurobiol.*), and it may be the case that a temporally precise within-trial urgency signal somehow relies specifically on recruitment of the former. Second, in addition to its potential reliance on a specific neural system for representing elapsed time, it is possible that the within-trial urgency that we report presently is partly or even exclusively generated by neural mechanisms that themselves only operate over relatively slow, supra-second timescales. However, in mitigation of both the above points, we note that urgency signals operating over much faster timescales have already been reported in the primate literature (Churchland et al., 2008, *Nat. Neurosci.*; Hanks et al., 2014, *eLife*).

We thank the reviewer for raising the pertinent question of why it is that we were able to observe time-dependent adjustments to the decision bound, while previous studies with human subjects were not. We agree that this question merits further discussion in the manuscript, and as such have now extended the associated part of the Discussion section (p.16-17). We now also explicitly state, in the Methods (p.22), our motivation behind the heavy punishment for missed deadlines.

On a related note, it is not very clear from the main text or SOM what would have led to the relatively long free-response RTs gathered in the current study. One possibility is the higher limit for premature responses. I think overall more information should be provided (preferably in the main text) regarding the key procedural details (e.g., average coherence, etc.).

The relatively long RTs in our free response condition are again likely a result of a combination of experimental design choices:

(1) As mentioned above, we employed a subject-specific difficulty calibration routine which ensured that the perceptual discriminations were set for all subjects to be relatively difficult (equating to 75% accurate responses in the DL condition). This difficulty translates, of course, to relatively slow RTs, especially under no external speed pressure.

(2) In a specific effort to minimize a possible tendency toward a time-dependent decision policy in free response blocks, we strongly emphasized to subjects that they should be as accurate as possible and, moreover, should try to reach the same level of confidence in their decision on every trial of this condition, irrespective of how long that decision took to be made. Our reasoning here was that the type of task instructions that are often provided on free response choice RT tasks ('be as fast and accurate as possible' or some variant thereof) are vague, open to individual differences in interpretation, and can be interpreted as encouraging time-dependency.

(3) As suggested by the reviewer, our somewhat high limit on premature responses (200ms) may have also shaped subjects' strategies by prioritizing greater response caution. (Note though that this criterion was not violated at all during task performance, extremely rarely during practice, and judging by our interaction with subjects, did not seem to exert a strong influence on task strategy.)

Again, we concur that the above details, and other methodological specifics, should be made clearer in the manuscript and have now added to the Methods (p.21-22) and Supplementary Methods (p.11-13) sections to reflect this.

See below for a more major issue:

There is a large difference between the reward attained for correct responses and penalty amount associated with missing the deadline. Frazier & Yu (2007) and Karsilar et al (2014) have formulated the optimal threshold collapse trajectories (as a function of trial time) for those cases in which there was no explicit penalty for missing the deadline. It is these conditions for which the optimal strategy is minimal threshold setting (i.e., meeting thresholds) at the deadline.

When there is a large relative penalty associated with missing the deadline (e.g., -10:1) and given endogenous timing uncertainty regarding the deadline, reward-rate maximization requires minimal threshold settings (e.g., threshold = $x(0)$) much earlier than the deadline. A statistical decision theoretic approach to the decision scenario at hand would reveal this relation. Consequently, under this account accuracy should converge on the chance level performance prior to the deadline. However, this is not what has been observed in the current work. Thus, when evaluated within the framework of optimality, it is not clear how the findings of the current study are different from the results of Karsilar et al (2014) - note that optimal threshold collapse trajectories should be different between these two studies given the payoffs. I think addressing this issue would strengthen the manuscript and contribute to its scope.

We think that this is an excellent point and we completely agree that greater consideration of the question of optimality in our specific task context would strengthen our manuscript. Accordingly, using our urgency drift diffusion model and assuming a fixed, representative combination of drift rate, drift rate variability, non-decision time and static urgency (taken as the group-mean fitted values for these parameters under deadline), we have now derived the optimal (that is, reward-maximizing) additive urgency signals under various levels of endogenous timing uncertainty, parameterized according to two closed-form functions: the three-parameter logistic function that we employed in the reported model fits, and a two-parameter simple linear function that imposes much greater constraint on the shape of the urgency signal but can closely approximate the shape of the fitted urgency signals derived from our data. This analysis is now described in the Results (p. 11) and Supplementary Methods (p.16), and the results are depicted in a new Supplementary Figure 3.

In brief, the reviewer's main point is clearly borne out by this analysis: Given the relatively severe penalty for missed deadlines on our task and assuming some degree of timing uncertainty, the decision policy that maximizes expected reward is in most cases to adopt a zero criterion on accumulated evidence (i.e. an urgency signal that reaches the decision bound) some time before the deadline actually arrives. In addition, the greater the timing uncertainty, the earlier this point of maximum urgency occurs. When the urgency signal is allowed to take the logistic form, its deflection toward the decision bound is relatively delayed (the latency of which is determined by the level of timing uncertainty), and steep – this is generally consistent with the previous observations of Frazier and Yu (2007) and Karsilar et al. (2014), and very different from the gradual, quasi-linear urgency that we observed in the model fits. By contrast, when the urgency signal is constrained to be linear in the optimality calculations, its trajectory under moderate-to-low timing uncertainty is in fact quite closely approximated by the original fits to data – including, interestingly, not fully reaching the decision bound by the time of the deadline. [This latter feature is present in the optimal case because the penalties associated with the very few missed deadlines (<0.1% of all trials) under such an urgency regime are offset by the greater reward gleaned from the shallower urgency signal prior to the deadline.]

A comparison of these reward-maximizing policy settings with those derived from our empirical data is interesting and informative in at least two respects. First, it suggests that although adaptive and time-variant, our subjects' policy adjustments under deadline were not strictly optimal. Our results are somewhat similar to those of Karsilar et al. in this deviation from optimality. However, we stress that a crucial difference remains: Whereas the patterns of behaviour associated with deadline-induced speed pressure in that study could be readily reproduced by a time-invariant sequential sampling model (the regular DDM with fixed decision bound and between-trial variability parameters), several features of our findings cannot (Figure 3g-h). Thus, our findings make the critical advance of highlighting that human decision-makers can

employ a time-dependent policy, albeit apparently not to the extent required for reward-maximizing performance in our task context.

Second, this analysis may serve to highlight a basic constraint on the form of time-dependency that decision-makers are capable of adopting. When we allowed the urgency signal a high degree of flexibility, its reward-maximizing form was a poor match to the urgency signals derived from model fits to our subjects' behaviour; however, when the reward-maximizing urgency was constrained to be strictly linear, the fitted signals approximated its trajectory under low timing uncertainty. Strikingly, the urgency signals derived from neural data and fitted to the behaviour of non-human primates are remarkably similar in form (Hanks et al., 2014, Figure 6a; Thura & Cisek, 2016, Figure 8b) to the linear urgency signals that we observed here (albeit operating over different timescales). Moreover, the shapes of our fitted urgency signals under deadline were highly similar across all 21 fitted subjects (Figure 3b), despite being afforded the same freedom of form as the optimality analyses with logistic urgency. Collectively, these findings may point toward fundamental constraints on the neural machinery that generates time-dependency in the decision process. We now raise this intriguing possibility, alongside the general issue of optimality, in the Discussion section of our manuscript (p.17-18).

There is not enough data at the critical region right before the deadline. One way to address this problem would have been shortening the deadline (although this has not paid off in other studies) or increase the number of data points.

Indeed, we acknowledge that a low number of trials immediately preceding the deadline is an important issue when attempting to estimate accuracy at this critical time point. We point out that low trial counts at this time point will be a common problem to studies dealing with a gradual as opposed to an abrupt time-variant adjustment in decision policy, as certainly appears to be the case in our data given the smooth right-tails of the RT distributions and the shapes of the fitted urgency signals: This is because the lower the evidence criterion gradually gets as the deadline approaches, the greater the probability that the criterial level at a previous time-point will have been met, such that the relative density of trials at the critical final time point will inevitably be extremely low. As such, one would need to collect an extremely large number of data points per participant to effect any substantive increase in the number of trials at this critical region: If one has ~5 trials per participant in the 50ms preceding the deadline, for example, the duration of the entire experiment would need to be doubled just to glean another 5 trials. We regret that, because we report data from a total of 91 subjects and each of the studies that we report also entailed some form of physiological measurement which in turn requires time and resources, longer per-subject measurement times were not feasible.

We acknowledge that this relative paucity of trials immediately preceding the deadline prohibits a precise characterization of the shape of the empirical CAF at this time point, and we now mention this caveat in the Methods section of the manuscript (p.23-24). Nonetheless, we emphasize two features of our approach that allowed us to arrive at what we believe to be a reasonable approximation. First, our single-trial logistic regression approach exploited consistencies in the temporal evolution of the conditional accuracy function (CAF) to provide an estimation of accuracy at this critical time-point that circumvented the fact that the low trial counts would lead to very noisy estimates if one were to, say, simply calculate average accuracy in a narrow time bin. The appropriateness of this approach relies on there not being an extreme, abrupt change in decision policy, which would skew the regression fits, but in our view this is a very reasonable assumption to make: again, the RT distributions speak against any abrupt change in policy, and previous primate studies (Hanks et al., 2014; Thura & Cisek, 2016) also presented data that favour gradual rather than abrupt changes. Second, the central tendency of the CAF in this critical region that is afforded by averaging across the entire cohort of subjects, as we did here, presumably further circumvents the signal-to-noise issues inherent in the low trial counts at the individual-subject level.

It is not clear if the RT distributions were based on pooled or vincentized data.

The plotted distributions are pooled, not vincentized. We have now made this clear in the legends to Figures 1 and 3.

Reviewer #2 (Remarks to the Author): *This manuscript describes several experiments on human perceptual choice (all using the random-dot motion discrimination task), that examine the putative influence of a time-dependent "urgency" signal. In experiment 1, subjects are tested in a Free Response (FR) condition as well as a Deadline (DL) condition, in which responses longer than 1.4s incur a heavy penalty (equal to -10x the value of a single correct choice). Behavior is analyzed using conditional accuracy functions (CAFs) showing a strong effect of urgency in the DL condition, consistent with an approximately linear urgency signal. EEG data in the 8-14Hz band shows: 1) reduced power in the DL vs. FR condition at baseline and at motion onset; 2) condition-independent power contralateral to the responding hand just prior to response onset; 3) an RT-dependent difference in the contrast between contra-vs-ipsi power in the DL condition. The authors suggest this reveals the presence of a time-dependent signal (especially in the DL condition) that is effector-independent, consistent with an urgency signal. In Exp 2, subjects perform almost the same task, but adjusted to allow measurement of pupillary responses. The data shows: 1) a significant increase in the baseline pupil diameter in the DL vs FR task; 2) a time-dependence on pupil size that is consistent with a linear ramp; 3) an RT-dependence of pupil size at response time in the DL condition. Both of these sets of results tell the same story: That in the FR task there is a flat urgency signal, but that signal becomes very steep in the DL task. Two kinds of computational models are applied to this data: a classic drift-diffusion model (DDM) with or without additive urgency, and a leaky competing accumulator (LCA) model with a time-dependent transfer function. Simulations show that a DDM without urgency cannot capture the behavioral data, but a DDM with urgency can, and the LCA with urgency can capture both the behavioral data as well as neural processes inferred from EEG and pupillary analyses. The take home message is that there exists a time-dependent urgency signal that modulates the gain of processing, especially when there is time pressure. Finally, the authors show that this generalizes to conditions with only mild time pressure. They re-analyze data from a previous study (Exp 3) and a new experiment (Exp 4) in which there was no heavy penalty for missed deadlines. Again, they use conditional accuracy functions to show a strong effect of urgency on behavior.*

Overall, I found this to be a very interesting manuscript presenting highly compelling data. Most impressive was the good qualitative match between urgency signals (in Exps 1 & 2) derived in three different ways: from behavior, from EEG, and from pupil diameter. This was strengthened by solid and thorough modeling, using an abstract formalism (DDM variants) and using a neural circuit model (LCA). In all cases the conclusions were congruent. Finally, the additional experiments (3 & 4) further show the generality of the conclusions to other situations. This provides a strong counter argument to recent papers claiming that urgency does not play a role in human perceptual choice.

Although I liked this paper very much, I have some suggestions for further improvement.

GENERAL COMMENTS:

1. As the authors are aware, a dropping conditional accuracy function (CAF) can be produced without any actual rising urgency. This is stated explicitly on line 548, in the methods, but it is not clearly acknowledged in the main body of the paper. This could create the impression for some readers that if the CAF is decreasing, then an urgency signal must be present. I think it's better to clarify, up-front, that a dropping CAF is expected in any paradigm in which subjects are allowed to determine their own response time - because attention/arousal fluctuates, trials in which it happened to be low will tend to have long RTs and also be less accurate, skewing the distribution to create a dropping CAF.

We thank the reviewer for pointing out this point of potential confusion and, as suggested, have now clarified in the Results section that a dropping CAF does not necessarily imply time-varying urgency (p.6).

The authors already allow for this effect by permitting between-trial variability in gain (in both DDM and LCA models), and their simulations show that this is enough to generate the mildly decreasing CAF in the FR condition but not the dramatically steeper one in the DL condition (Fig. 3h). Nevertheless, I think their data might make it possible to control for this confounding effect even more convincingly. In particular, in study 2, can't one use the baseline pupil diameter as an estimate of the attention/arousal

on a given trial? If so, then one could select out trials with a similar baseline pupil diameter and then repeat the analyses in Fig. 1. Any remaining negative slope of the CAF could then be more confidently attributed to urgency.

We think that this is a nice idea. We conducted the analysis recommended by the reviewer, using subsets of trials from the study 2 cohort that were precisely matched for baseline pupil diameter (via the same iterative matching procedure described for the RT-matching EEG analysis reported elsewhere in our manuscript) and comparing CAF slopes across DL and FR conditions. Even with trial matching, a significant difference in CAF slopes was observed ($p < 1 \times 10^{-6}$), such that the declining slope in the DL condition was steeper than that in the FR condition. We now briefly mention this analysis in the Results section (p.6) and report it fully in Supplementary Figure 1.

Note that although the CAFs depicted in the new figure are of very similar shape to those initially reported in Figure 1c, there is one salient difference: estimated accuracy at the deadline is above chance ($58.3 \pm 2.5\%$; one-sample t -test with $H_0 = 50\%$: $t_{22} = 3.4$, $p = 0.003$). This discrepancy may be due to two factors. First, the new trial-matching analysis is based on significantly fewer trials (mean = 307 per condition) than the original analysis (720 per condition), which equates to very few trials immediately preceding the deadline and, thus, less reliable estimates. Second, we opted for inter-leaved DL and FR blocks in study 2 which, despite our efforts with additional short practice blocks prior to each experimental block, may have interfered somewhat with subjects' abilities to appropriately adjust their decision policies between conditions.

A simulation of a DDM with fixed bound and no variability in gain (drift rate) would presumably not show a negative CAF in either condition.

This is correct. Indeed, we state in the Methods section (p.27) that given any combination of drift rate, decision bound, non-decision time and diffusion coefficient (s), the DDM without variability parameters will produce a completely flat CAF and thus cannot account for negative-going CAFs observed empirically.

2. Although the authors strongly argue against one of the assumptions of the DDM, the fixed bound, they still adhere to the assumption of perfect integration, at least in the text and DDM simulations. However, it has recently been argued that perfect integration does not occur, even in the random-dot motion discrimination task (Carland et al. 2016, J Neurophysiol), but that instead, the motion signal is processed with a low-pass filter with a time constant of less than 250ms. This translates to a large leak term in a model such as the LCA used here. As noted by Ditterich (2006), once one allows for time-dependent gain (urgency), data from motion tasks can be explained without perfect integration. This implies that the authors could also relax that assumption and simulate all of their data using models with a short time constant. The LCA model used here is in fact a "leak dominant" LCA that has a time constant determined by: $timestep / (\lambda - \alpha - \beta)$, or $0.01 / (0.201 - 0.110 - 0.073) = 555ms$ (assuming I got the math right). That is not compatible with the data of Carland et al. These parameter values were found by assuming a constant urgency $g = 0.01$ for the FR task. However, it would be interesting to see whether you can still fit the data from your FR and DL tasks if instead of assuming a constant g for the FR task, you constrain the time constant to be shorter than 250ms and let the other parameters vary, including the urgency signal. I understand that in these fitting exercises one does not want to let all parameters vary, and you've chosen to fix g for the FR task on the basis of your data. However, unless you claim that there is something different about your task and the task of Carland et al., you can instead place a constraint on the time constant and let g vary over time (This might produce a non-constant urgency in the FR task, but it might not: The strongly exponential right-tail of your RT distributions might in fact be explainable even if the time constant is short and urgency is flat - it would be something akin to a low-pass filter motion detector with a "survival function"). It would be worth trying, because with the same model you could probably also simulate all of the data from the Carland et al. study, and abandon the assumption of perfect integration.

Our initial modelling efforts adhere to the assumption of perfect integration for a specific and, we believe, well-founded reason. There is a rich history of behavioural modelling in the field of human decision-making research that has strictly adhered to this principle, and it is from this sphere that much of the criticism

surrounding the notions of time-dependency and urgency has emanated. Therefore, one of the most powerful arguments in favour of the existence of time-dependency would, in our view, be a demonstration that time-dependency is clearly necessary to fit an empirical dataset using the type of modelling framework that is commonly employed in this field. This is what we provide with our DDM analyses depicted in Figure 3.

As the reviewer alludes to, our second modelling framework (Figure 6) relaxes the adherence to perfect integration significantly; in fact, here we placed relatively little constraint on the timescale of accumulation (see below), and instead constrained the time-course of network gain in the FR condition in a manner that, importantly, was informed by our empirical pupillometric findings. In this case then, the model was free to adopt an effective time constant on the order of that reported by Carland et al. while simultaneously satisfying the constraint imposed by our pupil results, but such a configuration presumably led to a somewhat poorer fit during the fitting procedure. [Note that while the possible range of effective time constants was broad, it was limited by the following fitting constraints: leak should exceed recurrent excitation; recurrent excitation should exceed lateral inhibition by a fixed factor; and, there should be a lower bound on recurrent excitation in the fitting. The latter constraint we somehow failed to acknowledge in our initial submission (this has been corrected), but was necessary: without this constraint, the fitted recurrent excitation and lateral inhibition were typically low, and the model failed to produce competitive accumulation dynamics and the time-dependent increase in the common activation of both accumulators in the DL condition.]

Nonetheless, given the recent findings in the Carland et al. paper (which we had not seen by initial submission and thank the reviewer for pointing us toward) and other very relevant recent work from the Cisek lab, we do see the value in asking whether a network with a shorter time constant of accumulation and time-varying gain in both conditions is capable of generating the behaviour that we observed empirically. So, as requested by the reviewer, we attempted to address this question with additional model fits. First, to clarify, the reviewer is partially correct in their formulation of the time constant in our LCA model as $dt / (\lambda - \alpha - \beta)$. This would represent what we will call the 'effective differential leakage' – that is, the time constant of the difference in activation between the two accumulators – *without* both a non-linearity in the network transfer function and time-variance in the shape of this function. Under fixed low gain (as in our original fits to the FR condition) and without any lower bound on activation, we believe that this effective differential leakage is conceptually equivalent to the time constant of accumulation in Carland et al. and the two are directly comparable. With the lower bound and time-varying gain, however, the comparison is a lot less straightforward. So that we could at least attempt to address the reviewer's request, in our additional model fits we took $dt / (\lambda - \alpha - \beta)$ to be a *rough approximation* of the time constant at low network gain and proceeded from there. We stress, though, that the results of these analyses, and their relationship to those reported in Carland et al., should be viewed with this caveat in mind.

As a starting point in the additional model fits we enforced the constraint that, at the beginning of the decision process at least, network gain in the FR condition was low – accordingly, g at $t = 0$ in the FR condition was fixed at 0.01. Next, we fixed the approximated time constant [$dt / (\lambda - \alpha - \beta)$] to equal 167ms – identical to the time constant used in the Carland et al. study. Then, we fit the LCA model to the behavioural data from the FR and, subsequently, DL conditions using the same constraints and fitting procedures as for our original LCA fits, with the exception that the network gain was now allowed to vary as a function of elapsed decision time in both conditions. This version of the model also provided a reasonably good fit to the observed behaviour (with the slight exception of a poorer fit to the leading edge of the FR RT distribution relative to our initial fits, but a slightly better account of the CAF), and was similarly able to generate, qualitatively at least, the time-dependent increase in common activation of both accumulators in the DL condition. Remarkably, the fitted gain time-course for the FR condition remained flat for the majority of decision times, much like the completely constrained gain time-course in our initial fits, even though it was free to take a variety of other forms. In sum, it seems the reviewer's intuition was correct and the model can reproduce similar effects as in Figure 6 when the time constant matched that used in Carland et al. (2016). We now refer to this additional analysis in the Results section (p.14), describe it in the Supplementary Methods (p.17), and show the results in Supplementary Figure 6.

SPECIFIC COMMENTS:

Lines 77-87: I think more information should be provided in the main text on some of the details of the experiment. In particular, the payoff for correct choices (0.5c), the loss for errors (-0.5c) and the heavy penalty for missed deadlines (-5.0c) should be made clear. It would also be useful to explicitly state that the inter-trial interval was random within a range of 1.65-2.15s but that trials were not initiated on a rhythm (i.e. the timing of the next trial dependent at least in part on the RT in the previous trial). This is important because it motivates strategies for reward rate maximization.

We thank the reviewer for bringing each of these points to our attention have now included each of the specified details in the Methods section of the main text (p.21-22). We also better specify the extent of the large penalty for missed deadlines in the Results (p.5).

Line 140: Desynchronization is misspelled.

Following the advice of reviewer #3, we have now abandoned the term desynchronization entirely.

Line 171 and elsewhere: The term "competition" is used here to mean common activation in two accumulators. But many people use that term to imply a direct process of competition, such as mutual inhibition through recurrent inhibitory connections. Saying "stronger competition" has very different implications given these usages, which can be potentially very confusing. I would recommend using a different term. For example, here you could just explicitly say "common activation ... can provide a proxy". The term competition is used elsewhere as well, and you might want to consider changing it. Alternatively, you could use the term, here and elsewhere, as long as you define it more explicitly and distinguish it from a mutual inhibition process.

We agree completely. We have now replaced the various instances of the term 'competition' with 'common activation' or a variant thereof.

Line 190: The urgency functions were fitted with a logistic function. But it seems to me that you'd get almost as good a fit with a purely linear function, for both FR and DL conditions. Is there any real advantage of adding this extra complexity? Note that the initial rising CAF and inflection can be produced with just large noise that causes early guesses.

We favoured the logistic function because we did not have a strong *a priori* hypothesis about the shape of the urgency signal, and the logistic function is considerably less prescriptive than a purely linear function – it at least gives the fitting procedure the *opportunity* to produce urgency signals of a variety of different forms (including, importantly, the reward-maximizing form now illustrated in Supplementary Figure 3). The fact that the best-fitting form is near-linear is then itself instructive, as it tells us that an approximately linear urgency signal provides a better fit to the observed data than other candidate shapes. We also note that the increase in model complexity is not extreme: The logistic function adds only one extra parameter relative to the simple linear function (or rather, two when fitting to both DL and FR conditions and this extra parameter is allowed to vary), and does not penalize the urgency models to the extent that it meaningfully affects interpretation of the quantitative model fits (the urgency DDM BIC scores are very reliably lower than their regular DDM counterparts in our existing fits).

Line 267: I don't understand the meaning of the subscripts Onset-PM and Resp-PM.

These subscripts were intended to refer to the onset and response regressors in the extended pupil model described in this paragraph that were parametrically modulated by response time. We now make this more explicit in the text (p.13) and no longer use these subscripts.

Line 281: I assume that when you say "first cohort" you mean Exp 1. This should be explicit.

We have replaced this with "first experiment reported above".

Line 284: What does the word "latter" refer to?

This refers to the dynamics of the evidence accumulation process. We have now slightly rephrased to clarify.

Line 298: Not penalized in any way: Does this mean there were no monetary rewards or losses in Exp 3 and 4? This should be explicitly stated.

The incentive schemes for studies 3 and 4 were different: in study 3, subjects received a fixed payment that was not performance-dependent; in study 4, they received a fixed payment and additional monetary rewards for correct responses, but were not monetarily penalized for missed deadlines. In any case, we have now rephrased this line slightly such that it now reads “there was no explicit monetary penalty for missed deadlines”.

Line 315: What is meant by "cost of deliberation"? I think a more natural way of motivating a time-dependent urgency is to just assume that subjects want to maximize reward rate. That implies that time is in the denominator, thus automatically becoming a "cost". I'm not sure we need to add any other assumptions of deliberation costs.

With this phrasing we intended to allude to the work of Drugowitsch et al. (2012, *J. Neurosci.*), which indeed assumes that decision makers aim to maximize their reward rate, but that reward rate is partly determined by a time-varying cost function (their equation 1). When this cost function increases with elapsed decision time (due to several possible factors, such as the use of deadlines, mixed difficulty levels, etc.), the optimal policy is to adopt a decreasing criterion on accumulated evidence; but when it is flat, maximizing reward rate does not necessarily entail such time-dependency (see their Figure 3a). As such, we have retained our specification of a time-invariant policy being sub-optimal in cases where the cost function grows with time. We now refer to this as “the cost of continued evidence accumulation”, which is closer to the language used by Drugowitsch et al.

Lines 424-426: If you don't put the numbers on payoffs and penalties in lines 77-87, then they should at least be here.

We have opted to include this information in the Methods section specified by the reviewer (p.21-22), and now also further highlight the asymmetry in rewards/penalties early in the Results section (p.5).

Lines 427-428: Did subjects do all of these trials in a single daily session or were they spread over two or more days?

These trials were administered in a single daily session. We have now made this clear (p.22).

Line 432: Were the easy and difficult trials fully interleaved or run in blocks?

The different difficulty levels were fully inter-leaved, which we have now made clear (p.22).

Line 605: You can cite Standage et al. (2011) as implementing time-dependent gain in this way.

We agree and have now cited Standage et al. here.

Fig. 2: I suggest adding labels for "FR" and "DL" to panels C and D, so that it's clear at a glance. I would also suggest adding a y-scale to the DL data in panel D.

We have now incorporated each of these suggestions into Figure 2 (and what is now Supplementary Figure 8).

Fig. 4: Why are the analysis windows in panel C different for FR and DL? Maybe there's a reason, but otherwise it looks like cherry-picking. My guess is you could just use the same window for both.

We chose these windows because they are centred on the respective latencies of peak dilation in each condition (derived from the grand-average dilation waveforms in each condition), and we wanted to avoid potentially biasing the results by using an off-peak window in one or both conditions. To verify that the

reported effects are robust to the specific selection of temporal windows, we now report an additional analysis in which we run sample-by-sample linear regressions at each time-point across the entire onset- and response-aligned pupil dilation waveforms; the resulting time-series of regression coefficients chart the temporal evolution of the associated effects. This analysis, which we now refer to in the Results section (p.12) and depict in Supplementary Figure 4, shows that the reported effects, and in particular the critical Condition*RT interaction, are indeed highly robust.

Fig. 3 vs. Fig 6: In Figure 3b you show a DL urgency function with a decreasing slope, and in Fig. 6c it has an increasing slope. But why the complexity? I would expect that a purely linear function would get you a fit that is almost as good, in both cases.

Again, we did not wish to be overly prescriptive with respect to the possible shapes of the urgency signals. It should also be noted that the urgency signals in the two models are not completely comparable: in the DDM the urgency signal is additive and linear in the effect it has on the accumulation process, while in the LCA it is multiplicative and non-linear. Thus it is perhaps not too surprising that they vary somewhat in shape.

Supplemental materials: As noted above, I think many of the details of the experimental paradigms should be moved to the methods section in the main text. I found that I was confused about several important details, which could impact on interpretation, until I read the supplemental section.

We certainly take this point, apologize for the confusion, and have now made an effort to integrate more methodological details into the Methods section (p.21-22) of the main text. We humbly ask the reviewers to bear in mind, however, that we are subject to a relatively stringent word limit with this submission, and so by necessity have left a significant amount of finer detail in the Supplementary Materials. We hope that our main text now provides sufficient detail for the general reader to comprehend the critical aspects of our design and analysis approach, while experts interested in more specific details can still turn to the Supplementary Materials.

Reviewer #3 (Remarks to the Author): *The manuscript investigates the mechanisms underlying human decisions based on perceptual evidence. The authors focus on the question if the employed decision policy depends on time or not, and combine behavioral, EEG and pupilometric evidence with sophisticated modeling to show that it does, and that mechanisms modulating the gain on evidence integration are able to mimic the presented observations. The work thus presents an interesting and timely contribution to the discussion on the exact mechanisms underlying human decisions, an in particular acts as an important counterargument to a series of recent papers claiming that the human decision policy is time-independent. The time-dependence of such a policy has already been established for non-human primates, but the concise set of provided evidence showing this for human decision-makers is, to my knowledge, novel.*

My main concern about the current version of the manuscript is the over-interpretation of the EEG results. While EEG recordings give us some idea about the time-course and coarse localization of neural activity, using them to distinguish between collapsing bounds and an urgency signal seems far-fetched. There might be some non-linear mapping between neural activity and the EEG signals that produces similar measures if the system implements collapsing bounds. This is not to say the evidence does not support an urgency signal. I just think that the statements in the manuscript should be phrased more carefully. The same applies to the pupilometry measures, but in this case the authors are more careful in their statements ("plausible hypothesis").

Our focus on the specific EEG oscillatory signatures in question was motivated by a collection of recent studies that established the following critical links between these signals and the evidence accumulation process, most of which have direct analogues in properties of single neurons in areas of primate association cortex that have traditionally been associated with decision formation (e.g. area LIP): the incremental change in the power of these signals is strongly coupled with the weight-of-evidence afforded by individual samples during the 'weather prediction task' (Gould et al., 2012, *J. Neurosci.*; see also Wyart et al., 2012, *Neuron*); these signals exhibit a ramping time-course, the slope of which scales with the strength of sensory evidence on the RDM task (de Lange et al., 2013, *J. Neurosci.*); their build-up rate reflects the temporal integral of upstream neural signals that encode the sensory evidence (Donner et al., 2009, *Curr. Biol.*; O'Connell et al., 2012, *Nat. Neurosci.*); and, their baseline, pre-decisional levels are sensitive to manipulations of prior expectation in ways that are again consistent with encoding of the decision variable (de Lange et al., 2013, *J. Neurosci.*). This suite of findings establishes the EEG oscillatory signals in question as the best currently-available non-invasive proxies for the effector-specific neural evidence accumulation process, and provides the foundation for our current attempt to use these signals to elucidate the mechanistic basis of deadline-induced speed pressure. Moreover, we note that aspects of the reviewer's critique of EEG extend to almost all other functional measurement techniques currently available to human neuroscience research (including fMRI, MEG, ECoG, and invasively-recorded LFPs) that have also been used to make inferences about decision processes and evidence accumulation.

Nonetheless, we acknowledge that the relationship between the EEG signals at our disposal and the firing rates of decision-related neural populations is of course less direct relative to the single-cell recordings reported in seminal decision-making papers of the last few decades, and as a consequence, some of the more extreme statements in our manuscript should be tempered with this in mind. Accordingly, we have now attempted to be more careful, and generally used more tentative language, in our interpretation and discussion of the EEG results reported in our manuscript (p.7-9, 18).

A further concern about the EEG analysis is that the arguments in that section of the main text seem to be based on a model that is not elaborated on in the main text but rather only in Methods (and even there I don't find it particularly clear). In order to be able to follow these arguments, the authors should elaborate on the underlying model in the main text, maybe with a supporting figure panel?

We very much appreciate the reviewer's efforts to improve the clarity of our manuscript. In this case, we purposefully tried to avoid including a lot of exposition regarding the rationale behind some of the specific EEG analyses in the Results section: This section is already very long, we are constrained by a limit on word count, and we hoped that any readers who desired additional information about a given analysis (in particular the analysis of time-dependency in the EEG signals, which we suspect the reviewer is mostly

concerned with here) could turn to the Methods section for greater detail. (Indeed, the associated explanatory paragraph in the Methods section, p.25, lies at 249 words, which we remain convinced is too long for the Results and would disrupt the coherence of that section significantly.) We do, however, take the reviewer's point that we could do a better job of communicating our logic in this section, and have now attempted to do so. Specifically, rather than detail a specific model of the potential urgency signal at play (which we devote much of the remainder of the Results toward), we now draw specific analogies between our expected results and the previous reports on urgency signals in the single-unit primate literature (Hanks et al., 2014; Thura & Cisek, 2016; see revised paragraph of Results, p.9). We hope that this further clarifies our approach and helps to make our hypotheses more concrete to the reader.

Turning to interpretation of the pupilometry data, the authors suggest that "global gain" alone reproduces the effect of speed pressure on behavior and neural dynamics. In this context I wish that the authors could be more explicit in what they mean with "global gain", which, by itself, could mean anything. Surely, they don't mean that "gain" is cranked up everywhere in the brain, causing all the observed effects (and many more). In their model the "gain" is, in fact, fairly local. Thus, while "global gain" might sounds appealing, I think that it is too unspecific to make falsifiable predictions, and so should be revised.

By global gain modulation, we in fact do mean exactly that: a global, *brain-wide* change in the gain (or 'responsivity') of neural connections. We now include the term 'brain-wide' in the manuscript to further emphasize this (p.11). The concept of global gain modulation has quite a long history in the connectionist modelling literature (e.g. Servan-Schreiber et al., 1990, *Science*; Usher et al., 1999, *Science*) and has been specifically linked to ascending neuromodulatory systems (e.g. Aston-Jones & Cohen, 2005, *Ann. Rev. Neurosci.*); these systems project extremely diffusely (in the case of the locus coeruleus, to almost the entire cortex) and are very capable of modulating neural processing on a global scale. By association, as we outline in our manuscript, it has been suggested that pupil-linked arousal systems provide an excellent proxy for such global gain modulation. Indeed, recent studies have demonstrated via a combination of pupilometry, fMRI and psychopharmacology that fluctuations in pupil-linked neuromodulation are associated with truly brain-wide changes in functional connectivity profiles (e.g. Eldar et al., 2013, *Nat. Neurosci.*; van den Brink et al., 2016, *J. Neurosci.*; Warren et al., 2016, *J. Neurosci.*).

The reviewer states that the gain modulation in our model is local. This statement can be considered incorrect insofar as our gain modulation affects every connection that is included in the network (feedforward input, recurrent excitation, and lateral inhibition); but it can of course also be construed as correct in that we only model a specific (and very small) subset of all possible neural connections. An important point here is that global gain modulation as we, and several others, implement it exerts a multiplicative effect which enhances activation in task-relevant processing streams (i.e. those that are activated by the current task) but suppresses activation in irrelevant processing streams; thus whereas the gain modulation is global and brain-wide, its effective influence is very much shaped by the processes that the brain is engaged in at any moment in time. Although we have not tried, we suspect that incorporating other, task-irrelevant units and connections in our model and keeping the gain modulation truly global would not change the key effects of gain modulation on decision-related dynamics that we report here.

Additionally, we wish to reiterate a point that we make in the Discussion section of our manuscript (p.19): that a central appeal of global gain modulation as a potential mechanism for generating speed emphasis is that it can plausibly account for existing observations that speed emphasis appears to have a variety of 'extra-decisional' effects – including, as we report here, a small effect on non-decision time. While we do not attempt to formally model and reproduce such effects here, we think it will be very possible to do so via global gain modulation within the modelling framework we invoke. These effects would not be reproducible via strictly local modulation of the gain of evidence accumulation.

Lastly, we refute the claim that our implication of global gain modulation is unfalsifiable; rather, when it is specified in a computationally tractable and precise manner as here, it allows one to make very specific hypotheses about the effects of deadline-induced speed emphasis on behaviour and neural dynamics (for example, on brain-wide functional connectivity patterns as mentioned above) that can be tested in future studies.

When analyzing behavior the authors seem to suggest two models, only one of which is supported by the data (Fig. 1c). However, the description of the models is the main text (around I91) is not very clear. Is the first model supposed to behave like in the FR condition up to the deadline, at which point accumulation is interrupted and a decision is cast? If this is the case, the authors should also consider what happens if the deadline estimate is noisy (as will be the case) and conservative (biased away from the actual deadline). What would be the predictions of sharp cut-offs at noisy deadline estimates, and how would it compare to a model with a collapsing boundary? This should also be discussed in the context of the EEG / pupillometry analysis.

Again we appreciate the reviewers point, though again in this case we strongly favour keeping things more general at the point of the main text that the reviewer refers to (the second paragraph of the Results section). The scenario that the reviewer outlines – of an accumulation process that is abruptly interrupted at the estimated time of the deadline in the DL condition but otherwise identical across the two conditions – is certainly one plausible way in which decision-makers could react to deadline-induced speed pressure. But it is one of several ways that fall under the general mechanism of a changing criterion on accumulated evidence that we outline here and which, at this point in the manuscript, we have no strong preference for (alternatives include a more gradual rather than abrupt adjustment in the criterion, and a combination of static and time-dependent adjustments, the latter of which we later find support for). We are of the opinion that at this early point in the manuscript, it would be distracting to spend time outlining each of the possible ways that decision-makers could conceivably respond to the task manipulation – rather, our analysis of the empirical CAFs is aimed at arbitrating between two *general* accounts of how the decision process might be adjusted in response to speed pressure, and so we limit our explanation to these.

We stress that we believe that the distinctions raised by the reviewer with this point are certainly important, and indeed we go on to address these distinctions later in the manuscript. Toward the end of this first section of the Results, for example, we describe how our data do *not* appear to be consistent with an abrupt change in decision policy (p.6-7). Additionally, we have conducted a new analysis that devises what shape of urgency signal should be expected if our subjects were implementing policy adjustments in an optimal, reward-maximizing fashion, under different levels of noise (uncertainty) in their deadline estimates (Supplementary Figure 3). We do, however, have a strong preference for maintaining the current structure of the main text, wherein we begin with general distinctions between candidate mechanisms and only subsequently move to more specific distinctions.

Other points that would require additional justification are some of the choices made in the analysis of the various signals. For example, why was the pre-response μ measured as the mean power from -170ms to -50ms (I474)? This seems a fairly specific, but on the other hand rather arbitrary choice. Are the results robust to changes in these values? Discarding trials with $RT < 0.5s$ for EEG analyses (I482) also seems restrictive, particularly for the DL condition.

Each of the EEG analysis choices mentioned were made for a specific and principled reason. As we describe in the Methods section (p.24) and depict in Figure 2b (topography at middle inset), 8-14Hz power is subject to a strong decrease in the immediate post-stimulus period that is bi-lateral and generated occipitally but contaminates early ($< -0.4s$) portions of the motor preparation signals that were of paramount interest here, and it was of critical importance to mitigate the influence of this early occipital response on the reported effects. We did so by (a) restricting our analysis to the period immediately preceding response execution (in the temporal window that the reviewer points out, which was *not* arbitrary but rather centred on the latency of peak power decrease in the response-aligned grand-averages and purposefully chosen in a manner that was orthogonal to potential RT and condition*RT effects, as we now highlight in the manuscript of p.24; see Figure 2b, *right panel*); and (b) excluding trials with $RT < 0.5s$, in which the 8-14Hz power traces are dominated by this occipital response and would confound our analyses of these signals.

We also point out that discarding the small proportion of fast RT trials confers an additional advantage in that 0.5s is an appropriate cutoff for the so-called ‘fast guesses’ that are apparent in the left-side of the empirical CAFs (in the DL condition in particular; Figure 1c). These are assumed by some modellers to represent a distinct class of trials that are not reflective of the typical evidence accumulation process (e.g.

Ratcliff & Tuerlinckx, 2002, *Psychonom. Bull. Rev.*) and so might further contaminate our targeted analysis of urgency effects if left in the analysis.

How was the pupil response function time constant (1532) chosen? ANy of those and similar choices warrant justification.

The parameters for the pupil impulse response function were chosen to match the best-fitting parameters from the paper that originally described this function (Hoeks & Levelt, 1993) and which have been employed in other studies since (de Gee et al., 2014; Lempert et al., 2015). We apologize for not making this explicit in the manuscript and have now done so (p.27).

Prompted by our consideration of the reviewer's point, we also acknowledge that it is possible that these function parameters do not generalize to experimental settings other than those examined by Hoeks and Levelt, and that different parameter sets could conceivably produce different relative goodness-of-fits to the observed pupil time-series. Therefore, to examine the robustness of the critical effects that we report in Figure 5b,c, we repeated the associated analyses for a wide variety of different shapes of impulse response function. As is now depicted in Supplementary Figure 5, this extended analysis revealed that the initially reported effects are indeed highly robust: for the DL condition, the model in which the sustained temporal component took the form of a linear up-ramp was the best-fitting model for every combination of parameters examined; for the FR condition, the model in which this component was a boxcar was the best-fitting model for the large majority of parameter combinations (with the exception of a small subset of impulse response functions that were narrow and peaked late).

Minor comments:

166: "recent empirical and model comparison reports have reinforced this stance" - another recent one to cite would be Voskuilen et al. (2016): "Comparing fixed and collapsing boundary versions of the diffusion model"

We thank the reviewer for pointing us toward this recent and very relevant paper, which we had not seen, and have now cited it where recommended (p.4) and elsewhere (p.16, 17).

169: "combination of static and time-dependent urgency" - it is unclear at this point what static and time-dependent mean. The same applies to 1121, 1138, etc.

We now introduce the term 'static' in the second paragraph of the Introduction section (p.3), and have reformulated the sentence in question in a way that better fits with the terminology used in the previous paragraph (in which we do describe 'time-dependency' or a variant thereof several times, but not the specific phrase 'time-dependent urgency').

170: "urgency to optimally adapt to deadline-induced speed pressure" - optimality is never established.

We completely agree that this term is not appropriate, particularly in light of our new analysis of the optimal (reward-maximizing) shape of urgency signal, and have now removed it. We also point the reviewer to our extended response to reviewer #1's points about optimality, and to the associated Supplementary Figure 3.

199: "negative-going" (and later occurrences) - isn't there a better way to express this?

Instead of 'negative-going', we now use the terms 'decreasing' or 'negative CAF slope' where appropriate.

1103: Fig. 1c and Methods describe a piecewise linear fit to the CAFs, but the main text only reports a single slope. If results in the main text come from standard logistic regression over the full RT range, then these fits should also be shown in Fig. 1c. The same applies to later uses of piece-wise linear fits.

The reported slopes are indeed from the piece-wise fits, which we recognize was not properly communicated in the text. We have now amended our description of the CAF results to make this clear.

I135: "desynchronization" - unexplained mu wave jargon that should be avoided to make the manuscript accessible for a larger audience. Why not just call it a drop in mu power throughout the manuscript (on the side/in general: such a drop can be caused by other effects than desynchronization, such that calling it such might be misleading).

We agree and thank the reviewer for pointing this out. We have now rephrased to 'decrease in mu power' or some variant thereof throughout.

I533: which definition of BIC was used here? The one shown on I533 uses different terms than the one I am used to (or that is, for example, on Wikipedia).

This BIC expression is the expression used for calculating the BIC of models estimated via least squares (which was the case for the linear regressions that formed the basis of our pupil modelling analysis) – in this case, the likelihood term is expressed in terms of the residual sum of squares. We now make this clear in the manuscript (p.27).

We also need to clarify that this was *not* the expression used for calculating BIC scores for our diffusion model fits; rather, here we used the more familiar equation, $BIC = 2L + k \log n$. We had this in an early draft of the manuscript but it somehow got removed before manuscript submission. We have now added this back in (Supplementary Methods, p.15) and thank the reviewer for drawing our attention to the omission.

SI I272: was the $\log f(RT_i | \theta)$ split into correct/incorrect choices, or did you only model the reaction times?

The defective probability density f does indeed distinguish between correct and incorrect responses. We realise that the original expression in Supplementary Equation 7 did not clearly reflect this, so we have revised this and the associated text accordingly.

REVIEWERS' COMMENTS:

Reviewer #1 (Remarks to the Author):

Authors have addressed all the issues that I have raised during the previous round of reviews. The addition of the optimality analysis has extended the scope of the paper's coverage and tied it nicely with other relevant work. I think the paper will make a valuable contribution to the literature and I will be happy to see such nice dataset published.

Reviewer #2 (Remarks to the Author):

The authors have considerably improved the manuscript and responded very well to all of my comments and critiques. In particular, I was impressed with the new analyses they introduced to respond to my comments (Supplemental Fig 1 and 6), as well as the other new analyses stimulated by the comments of other reviewers (especially Supplemental Fig 3 and the associated discussion). I have only two remaining suggestions, both of which would require only minor modifications.

First, I think it would be useful to more strongly emphasize the question of time constants, which at this time is only mentioned briefly in the main text. I agree that demonstrating the usefulness of collapsing bounds within the "leakless" DDM framework is valuable and persuasive because that framework is so commonly assumed. But it is equally valuable to show that other common assumptions of that framework are not necessarily well-justified, despite their long history. The demonstration that the major findings of this study could be reproduced with a short time constant model is important, especially since in some ways the fit is better than the fit of the leakless DDM. I'm not advocating that the authors get into a debate about time constants, which their experiments were not designed to test. However, I think it would be useful to put the results shown in Supplemental Figure 6 into the main text. They could appear as additional panels added to (main) Figure 6, making it possible to directly compare the performance of the two kinds of models side-by-side. This would leave it for the readers to decide whether the commonly-made assumption of leakless integration is really necessary to explain these kinds of experiments.

Second, I don't see why the authors want to make the assumption, proposed by Drugowitsch et al. (2012) that there is an extra "cost of continued evidence accumulation", beyond the simple cost of time. Drugowitsch et al. showed (their Figure 3b) that even if the cost of accumulation is constant, the optimal bound for maximizing reward rate should be collapsing, as long as there is a mixture of trial types. In fact, the constant bound shown in Figure 3a only obtains because in the special case where every trial is strictly identical, maximal reward rates produced by different policies all intersect at a single point in time-vs-accuracy-space, and it doesn't matter how the bound reaches that point. In the present experiment, the stimulus coherence was fixed on each trial, but it is not likely that the subjects experienced each trial as strictly identical because there are always internal fluctuations in attention, arousal, etc. This is not really a major point, and the authors may ignore it if they so choose, but I think it is always better to avoid making assumptions that

are not really necessary.

Reviewer #3 (Remarks to the Author):

By clarifying some of the text, performing more extensive model comparisons, and by better justifying their parameter/model choices, the authors have significantly improved the manuscript, and addressed all my concerns.

I also welcome the additional analysis of what the optimal (reward-maximizing) urgency signal might be under different uncertainties of the deadline (the new Supp. Fig. 3), even though such analysis is not central to the message of the manuscript. There are two parts of this analysis that are still a bit unclear to me. The authors call the identified urgency signals "optimal", even though - to my understanding - they only investigate two parametric families of urgency signals: the logistic and the linear one. Thus, they only identify optimal parameters within these families, but there might be another parametric urgency shape that exceeds this "optimal" reward. Furthermore, they claim that the rapid rise of a logistic urgency signal is necessary to achieve the optimal reward. This might indeed be the case, but how do we know that a less rapid rise (e.g. quadratic) would not achieve a comparable or even higher reward? I don't believe that those considerations would change any of the conclusions of the manuscript, but it might be worth clarifying them.

We thank the reviewers and editor for taking the time to examine our revised manuscript and are extremely pleased that our revisions have been received positively. Please find our responses to each of the new comments from reviewers 2 and 3 below, along with descriptions of new amendments to the manuscript in light of these.

Reviewer #2 (Remarks to the Author): *The authors have considerably improved the manuscript and responded very well to all of my comments and critiques. In particular, I was impressed with the new analyses they introduced to respond to my comments (Supplemental Fig 1 and 6), as well as the other new analyses stimulated by the comments of other reviewers (especially Supplemental Fig 3 and the associated discussion). I have only two remaining suggestions, both of which would require only minor modifications.*

First, I think it would be useful to more strongly emphasize the question of time constants, which at this time is only mentioned briefly in the main text. I agree that demonstrating the usefulness of collapsing bounds within the “leakless” DDM framework is valuable and persuasive because that framework is so commonly assumed. But it is equally valuable to show that other common assumptions of that framework are not necessarily well-justified, despite their long history. The demonstration that the major findings of this study could be reproduced with a short time constant model is important, especially since in some ways the fit is better than the fit of the leakless DDM. I’m not advocating that the authors get into a debate about time constants, which their experiments were not designed to test. However, I think it would be useful to put the results shown in Supplemental Figure 6 into the main text. They could appear as additional panels added to (main) Figure 6, making it possible to directly compare the performance of the two kinds of models side-by-side. This would leave it for the readers to decide whether the commonly-made assumption of leakless integration is really necessary to explain these kinds of experiments.

We think that the reviewer’s point is fair. In response, we have now incorporated Supplementary Figure 6 into Figure 6 of the main text, as advised. We also now include a somewhat more detailed outline of the rationale and significance of this modelling exercise in the Results section (p.15).

Second, I don’t see why the authors want to make the assumption, proposed by Drugowitsch et al. (2012) that there is an extra “cost of continued evidence accumulation”, beyond the simple cost of time. Drugowitsch et al. showed (their Figure 3b) that even if the cost of accumulation is constant, the optimal bound for maximizing reward rate should be collapsing, as long as there is a mixture of trial types. In fact, the constant bound shown in Figure 3a only obtains because in the special case where every trial is strictly identical, maximal reward rates produced by different policies all intersect at a single point in time-vs-accuracy-space, and it doesn’t matter how the bound reaches that point. In the present experiment, the stimulus coherence was fixed on each trial, but it is not likely that the subjects experienced each trial as strictly identical because there are always internal fluctuations in attention, arousal, etc. This is not really a major point, and the authors may ignore it if they so choose, but I think it is always better to avoid making assumptions that are not really necessary.

There may be a small misunderstanding on this point. We are not assuming that in our case there is necessarily an “extra” cost of continued evidence accumulation, in addition to an assumed cost of time. Rather, we assume, as proposed by Drugowitsch and colleagues, that there is some cost function that prescribes the appropriate level of criterion change – what we refer to as ‘the cost of continued accumulation’ – and that the cost of time is a critical (and indeed, in our case, possibly the only) determinant of this cost function. When viewed in these terms, the above statement from the reviewer that “even if the cost of accumulation is constant, the optimal bound for maximizing reward rate should be collapsing, as long as there is a mixture of trial types” does not make sense, because the very fact that there is a mixture of trial types ensures that the cost function is not flat, and hence the cost of accumulation *cannot* be constant. In our manuscript, we mention this relatively abstract notion of the cost of accumulation only twice (p.4 and p.17), both in sentences in the Introduction and Discussion where we are writing in very general terms about contexts (not limited to our own) in which a collapsing criterion is appropriate and the cost function may be partly determined by factors other than elapsed time. To maintain this generality at these points of the manuscript, we prefer to keep the phrasing as it is currently.

Reviewer #3 (Remarks to the Author): *By clarifying some of the text, performing more extensive model comparisons, and by better justifying their parameter/model choices, the authors have significantly improved the manuscript, and addressed all my concerns.*

I also welcome the additional analysis of what the optimal (reward-maximizing) urgency signal might be under different uncertainties of the deadline (the new Supp. Fig. 3), even though such analysis is not central to the message of the manuscript. There are two parts of this analysis that are still a bit unclear to me. The authors call the identified urgency signals "optimal", even though - to my understanding - they only investigate two parametric families of urgency signals: the logistic and the linear one. Thus, they only identify optimal parameters within these families, but there might be another parametric urgency shape that exceeds this "optimal" reward. Furthermore, they claim that the rapid rise of a logistic urgency signal is necessary to achieve the optimal reward. This might indeed be the case, but how do we know that a less rapid rise (e.g. quadratic) would not achieve a comparable or even higher reward? I don't believe that those considerations would change any of the conclusions of the manuscript, but it might be worth clarifying them.

We thank the reviewer for highlighting the lack of clarity around some aspects of our reporting of the analysis depicted in Supplementary Fig. 3. The reviewer is indeed correct that the reward-maximizing signals identified by this analysis are only strictly optimal within the limits of the two closed-form functions that we employed (the logistic and simple linear functions), and that more flexible forms of urgency may yield (slight) increases in net expected reward. However, we wish to emphasise two points about our analysis that will hopefully better clarify why it is still highly informative.

First, we suspect that the reviewer understands this point but to be explicit: we certainly are not claiming that the reward-maximizing linear urgency signal is optimal across all possible forms of urgency – rather, it is only optimal within the limits of the simple linear function, and the less constrained logistic function provides a much better approximation of the truly optimal form. The value of the linear function analysis lies in highlighting that our actual subjects' urgency signals do appear to approach 'optimality' if we assume that their mechanisms for generating urgency are limited to only operating gradually in a manner that appears linear in time.

Second, although it is closed-form, the logistic function nonetheless allows us to achieve what is likely a very good approximation of the true optimal form of urgency. The logistic function is highly flexible, which means that it is capable of exhibiting not only the rapid deflection that we identify as reward-maximizing, but also more gradual time-courses that the reviewer alludes to in their second point; however, such gradual forms simply led to lower net expected rewards and hence, to answer the reviewer's question, we can be sure that they are sub-optimal relative to the rapid deflection form. Importantly, the reward-maximizing logistic signals identified by our analysis are also very similar in shape to the optimal forms identified by more complex and, critically, unconstrained dynamic programming analyses of time-varying decision criteria under deadline reported in Frazier & Yu (2008). Hence, we believe it is appropriate to state that our closed-form logistic function analysis allows us to *approximate* the truly optimal form of urgency given the representative set of remaining model parameters chosen.

We certainly acknowledge that some of the above points were not made sufficiently clear in our previous draft, and have attempted to address this in two ways. First, we have changed our description of the analysis in the main text (p.12) to make it clear that we are only approximating the optimal urgency signal with this analysis and are doing so using the closed-form logistic function in Equation 4 (neither of which were clear in the previous version). Second, we now provide some discussion of the distinction between closed-form and unconstrained optimality in the part of the Supplementary Methods where we fully describe this analysis (p.14/15 of the supplementary document). In particular, we now highlight the two key points regarding the flexibility of the logistic

function and the close match between the optimal logistic signals and the unconstrained signals identified in Frazier & Yu (2008), both of which suggest that the closed-form logistic analysis provides a good approximation of the true optimal urgency signal.